## Registered report

psychology

mate preferences, culture, attractiveness, social status

**Author for correspondence:**
Benedict C. Jones
e-mail: ben.jones@glasgow.ac.uk

# Chinese and UK participants' preferences for physical attractiveness and social status in potential mates

Lingshan Zhang[1], Hongyi Wang[2], Anthony J. Lee[3], Lisa M. DeBruine[1] and Benedict C. Jones[1]

[1]Institute of Neuroscience and Psychology, University of Glasgow, Glasgow, UK
[2]School of Psychology and Cognitive Science, East China Normal University, Shanghai, People's Republic of China
[3]Division of Psychology, University of Stirling, Scotland, UK

 AJL, 0000-0001-8288-3393; LMD, 0000-0002-7523-5539; BCJ, 0000-0001-7777-0220

Men are hypothesized to show stronger preferences for physical attractiveness in potential mates than women are, particularly when assessing the attractiveness of potential mates for short-term relationships. By contrast, women are thought to show stronger preferences for social status in potential mates than men are, particularly when assessing the attractiveness of potential mates for long-term relationships. These mate-preference sex differences are often claimed to be 'universal' (i.e. stable across cultures). Consequently, we used an established 'budget-allocation' task to investigate Chinese and UK participants' preferences for physical attractiveness and social status in potential mates. Confirmatory analyses replicated these sex differences in both samples, consistent with the suggestion that they occur in diverse cultures. However, confirmatory analyses also showed that Chinese women had stronger preferences for social status than UK women did, suggesting cultural differences in the magnitude of mate-preference sex differences can also occur.

## 1. Introduction

Two key factors are thought to drive sex differences in human mate preferences [1–3]. First, because fertility declines faster with age and requires a larger physiological cost for women than men, men are hypothesized to show stronger preferences for physical cues of reproductive capacity (e.g. youth, health and good nutritional status) in women than women do when assessing the attractiveness of potential mates [1]. Second, women bear greater costs of obligatory parental investment (i.e. pregnancy and

lactation) than men do, meaning they have both a greater need for resources and reduced ability to obtain resources [2]. Consequently, women are hypothesized to show stronger preferences for cues of the capability to invest resources in offspring when assessing men's attractiveness as long-term partners [1,3]. When assessing men's attractiveness as short-term partners, however, resources are thought to be less important and women are hypothesized to prefer men displaying cues that they are in good physical condition and will father healthy children [4]. Consistent with these hypotheses (see [5] for additional theoretical perspectives), studies have reported that women place greater emphasis on social status (i.e. resources) and men place greater emphasis on physical attractiveness when assessing potential long-term partners, while both men and women place great emphasis on physical attractiveness when assessing potential short-term partners (reviewed in [6]).

Because biological universals (i.e. sex differences in age-related decline in fertility and costs of pregnancy and lactation) are thought to underpin the sex differences in mate preferences described above, researchers have hypothesized that they should occur across cultures [1,3,6]. Indeed, there is good evidence that these mate-preference sex differences do occur in diverse cultures, at least when people express preferences for long-term relationships, such as marriage (e.g. [7]). While evidence for cross-cultural similarity in mate-preference sex differences for long-term relationships is well established, fewer studies have investigated mate-preference sex differences for short-term relationships. Moreover, studies investigating cross-cultural similarity in mate-preference sex differences have typically done so using either trait-rating or trait-ranking paradigms. These paradigms can be problematic because trait ratings do not require participants to trade off traits against each other and because trait rankings do not contain information about the relative strength of preference for traits [6,8].

Li *et al*. [8] developed the budget-allocation task to address the methodological limitations of trait-rating and trait-ranking paradigms. In the budget-allocation task, participants allocate a sum from a maximum total budget of 100 mate dollars to each of the following traits in a potential partner; physical attractiveness, social status, creativity, kindness and liveliness. Each participant performs this task twice; once when choosing for a long-term (marriage) partner and once when choosing for a short-term (casual sex) partner. Importantly, the budget-allocation task directly addresses the limitations of the trait-rating and trait-ranking tasks described above. Note that allocations represent participants' trait priorities, as well as their trait preferences.

To test for the hypothesized cross-cultural similarities in mate-preference sex differences, Li *et al*. [6] administered their budget-allocation task to US and Singaporean participants. Men allocated significantly more mate dollars to physical attractiveness than women did in both the US and Singaporean samples. Contrary to theory [3,5], this sex difference in preference for physical attractiveness was particularly pronounced when participants were choosing for potential short-term partners. By contrast, women allocated significantly more mate dollars to social status than men did in both the US and Singaporean samples. This sex difference in preference for social status was particularly pronounced when participants were choosing for potential long-term partners. Intriguingly, when choosing for potential long-term partners, Singaporean women allocated significantly more mate dollars to social status than US women did. Li *et al*. [6] suggested this latter result was consistent with the social status being more important for social interactions generally in Eastern than Western cultures [9]. It is unclear, however, why this cultural difference in preference for social status was only evident in women's preferences.

We will test for further evidence of these cross-cultural similarities and differences in mate-preference sex differences. We will use Li *et al*.'s [6] budget-allocation task to compare the UK and Chinese participants' preferences for physical attractiveness and social status in hypothetical short-term (casual sex) and long-term (marriage) partners.

In the current study, we will attempt to replicate three key results from Li *et al*. [6]. By contrast with Li *et al*. [6], who reported these results for US and Singaporean participants, we will attempt to replicate their key results in the UK and Chinese participants.

## 1.1. Prediction 1

Men will allocate significantly more mate dollars to physical attractiveness than women in both the UK and Chinese samples (Prediction 1a) and this sex difference will be significantly more pronounced when choosing for potential short-term partners than long-term partners (Prediction 1b). Note that, although Prediction 1b is what was reported in Li *et al*. [6], it is arguably inconsistent with theory [3,5].

**Figure 1.** Screen grab of interface used for the English-language version of the budget-allocation task.

## 1.2. Prediction 2

Women will allocate significantly more mate dollars to social status than men in both the UK and Chinese samples (Prediction 2a) and this sex difference will be significantly more pronounced when choosing for potential long-term partners than short-term partners (Prediction 2b).

## 1.3. Prediction 3

When choosing for potential long-term partners, Chinese women will allocate significantly more mate dollars to social status than UK women will. Note that, although Prediction 3 is what was reported in Li et al. [6], it is unclear why this cultural difference was not also observed for men.

# 2. Methods

The date of principle acceptance for this work was 16th October 2018. The accepted protocol is archived at https://psyarxiv.com/sybp4/ (version one). Data and analysis code are archived at https://osf.io/rkstx/.

## 2.1. Participants

We planned to test 125 heterosexual UK men and 125 heterosexual UK women at the University of Glasgow and 125 heterosexual Chinese men and 125 heterosexual Chinese women at East China Normal University (Shanghai). Only participants between the ages of 16 and 30 years of age born in either China (Chinese participants) or the UK (UK participants) were recruited. All procedures have been approved by the University of Glasgow, School of Psychology, Ethics Committee. All participants provided informed consent. Other than age, we collected no further demographic information from participants. Due to miscommunication among the researchers collecting data, we actually tested 132 heterosexual UK men and 127 heterosexual UK women at the University of Glasgow and 172 heterosexual Chinese men and 153 heterosexual Chinese women at East China Normal University (Shanghai).

## 2.2. Procedure

Each participant completed Li et al.'s [6] budget-allocation task. In this task, participants are instructed to distribute a total budget of 100 mate dollars across each of the following traits to choose a hypothetical partner; physical attractiveness, social status (i.e. good financial resources), creativity, kindness and liveliness. Each participant performed this task twice; once when choosing for a long-term (marriage) partner and once when choosing for a short-term (casual sex) partner. The order in which participants chose for long- and short-term partners was fully randomized and trait order was also fully randomized. On-screen instructions informed participants that each dollar corresponds to a percentile point on that trait. Instructions were presented in English for UK participants and Mandarin for Chinese participants. Data for traits other than attractiveness and social status are reported in an Exploratory analyses section. These traits were only included in the study because they were included in Li et al. [6]. Figure 1 shows a screen grab of the interface that was used for the English-language version of the budget-allocation task.

To ensure Mandarin translations accurately capture the nuance of the English terms used in the budget-allocation task, we followed the Psychological Science Accelerator's translation procedure (see Translation procedures section, below).

After completing the budget-allocation task, participants were asked to complete a manipulation-check task to ensure they understood what each trait represented (see Data exclusions section, below) and to report the age of their ideal long-term and short-term partner. These age-preference data are used in exploratory analyses testing for cultural and sex differences in age preferences.

# 3. Translation procedures

The Psychological Science Accelerator [10,11] has developed formal procedures for ensuring that instructions translated from one language to another accurately capture the nuance of the terms used in the original instructions [12]. This process reflects and extends the best practice in translating for cross-cultural research, as described in Brislin [13].

## 3.1. Translation personnel

Language Coordinator: Coordinated translation process and discussed the final version with translators.

'A' Translators: Translated from English to Mandarin and discussed the final version with coordinator and B Translators ($N = 2$, both bilingual)

'B' Translators: Translated from Mandarin to English and discussed the final version with the coordinator and A Translators ($N = 2$, both bilingual).

External Readers: Read materials for final clarity check ($N = 10$, all non-academics).

## 3.2. Translation process

Step 1 (Translation). Original document translated from English to Mandarin by A Translators resulting in document Version A.

Step 2 (Back-translation). Version A translated back from Mandarin to English by B Translators independently resulting in Version B.

Step 3 (Discussion). Version A and B discussed among translators and the language coordinator, discrepancies in Version A and B detected and solutions discussed. Version C created.

Step 4 (External readings). Version C tested on 10 non-academics fluent in the target language. Members of the fluent group asked how they perceived and understood the translation and agreed on three synonyms for each trait tested. Possible misunderstandings noted and again discussed as in Step 3. A group of 10 native English speakers also asked to agree on three synonyms for each trait to be tested. Note that the Psychological Science Accelerator's procedures for translation use two, rather than 10, bilingual speakers in Step 4.

This process produced the Final Translated Document, containing the instructions used in the study.

# 4. Confirmatory analyses plans

Analysis code (in R) for each analysis is available at https://osf.io/rkstx/ and in our electronic supplementary material. Only data for physical attractiveness and social status were used in our confirmatory analyses.

## 4.1. Analysis plan for Prediction 1

The amount of mate dollars allocated to physical attractiveness was the dependent variable in these analyses, which included data from both male and female participants. Prediction 1 was tested using separate ANOVAs for Chinese and UK participants' responses. Both ANOVAs had the between-subject factor participant sex (male, female) and the within-subject factor relationship type (marriage, casual sex). Prediction 1a will be supported if there is a significant main effect of participant sex, whereby men allocated significantly more mate dollars to physical attractiveness than did women in both the Chinese and UK participants' data. Prediction 1b will be supported by an interaction between participant sex and relationship type, whereby the effect of participant sex is significant in

both the casual sex and marriage conditions, but significantly greater in the casual sex condition than in the marriage condition, in both the Chinese and UK participants' data.

Power analyses (using G*Power 3.1) indicated we would have 90% power to detect effect sizes ($f$) of 0.15 for the main effect of participant sex (Prediction 1a) and 0.15 for the interaction between participant sex and relationship type (Prediction 1b), given 125 participants per group and a correlation between the repeated measures of 0.1.

## 4.2. Analysis plan for Prediction 2

The amount of mate dollars allocated to social status was the dependent variable in these analyses, which included data from both male and female participants. Prediction 2 was tested using separate ANOVAs for Chinese and UK participants' responses. Both ANOVAs had the between-subject factor participant sex (male, female) and the within-subject factor relationship type (marriage, casual sex). Prediction 2a will be supported if there is a significant main effect of participant sex, whereby women allocated significantly more mate dollars to social status than did men in both the Chinese and UK participants' data. Prediction 2b will be supported by an interaction between participant sex and relationship type, whereby the effect of participant sex is significant in both the casual sex and marriage conditions, but significantly greater in the marriage condition than in the casual sex condition in both the Chinese and UK participants' data.

Power analyses (using G*Power 3.1) indicated we would have 90% power to detect effect sizes ($f$) of 0.15 for the main effect of participant sex (Prediction 2a) and 0.15 for the interaction between participant sex and relationship type (Prediction 2b), given 125 participants per group and a correlation between the repeated measures of 0.1.

## 4.3. Analysis plan for Prediction 3

The amount of mate dollars allocated to social status for long-term relationships was the dependent variable in this analysis. This analysis included data from women only. Prediction 3 was tested using an ANOVA with the between-subject factor geographical region (China, UK) and the within-subject factor relationship type (marriage, casual sex). Prediction 3 will be supported if there is a significant main effect of geographical region, whereby Chinese women allocated significantly more mate dollars to social status than did UK women.

Power analysis (using G*Power 3.1) indicated we had 90% power to detect an effect size ($f$) of 0.15 for the main effect of geographical region (Prediction 3), given 125 participants per group and a correlation between the repeated measures of 0.1.

## 4.4. Data exclusions

Responses more than three standard deviations from the mean for that sex and dependent variable were excluded from the dataset prior to analyses. Specifically, we calculated the means and standard deviations for attractiveness and status allocations separately for men and women, then excluded from all analyses any participant who had at least one value more than three standard deviations above or below the sex-specific mean for attractiveness or status allocation. At Step 4 of the translation process, the external speakers were asked to agree on synonyms for each of the traits tested. Participants were asked to match these synonyms to the traits at the end of the study. Participants who failed this manipulation-check task for any trait were excluded from the analyses. No other exclusion criteria were applied.

# 5. Exploratory analyses

Data for traits other than attractiveness and social status are reported in an exploratory analyses section. These traits were only included in the study because they were included in Li *et al*. [6]. Exploratory analyses testing whether women value physical attractiveness more than other traits for short-term, but not long-term, relationships, while men value physical attractiveness more than other traits for both short- and long-term relationships are also reported in this section, along with exploratory analyses testing for cultural and sex differences in age preferences [5].

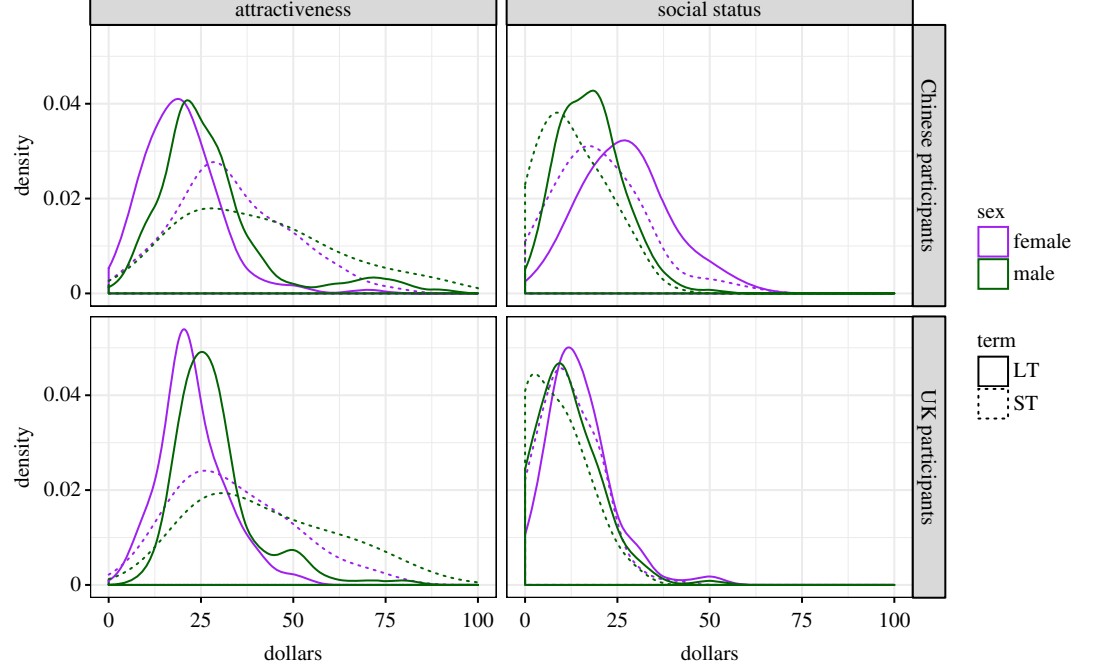

**Figure 2.** Distributions of scores used in our confirmatory analyses.

**Table 1.** Mean mate dollars allocated to physical attractiveness by group (and s.d.).

| sample | relationship context | male participants | female participants |
|---|---|---|---|
| Chinese | long-term | 27.63 (15.83) | 19.52 (10.23) |
| Chinese | short-term | 40.45 (20.86) | 33.26 (15.29) |
| UK | long-term | 28.90 (11.21) | 22.78 (8.56) |
| UK | short-term | 42.50 (18.74) | 33.64 (15.17) |

# 6. Results of confirmatory analyses

Distributions of the scores used in our confirmatory analyses are shown in figure 2.

## 6.1. Results of tests of Prediction 1

After data exclusions, 120 Chinese women, 142 Chinese men, 99 UK women and 113 UK men could be included in the final analyses of physical attractiveness.

In both samples, the main effects of participant sex (Chinese: $F_{1,260} = 20.28$, $p < 0.001$; UK: $F_{1,210} = 21.09$, $p < .001$) and relationship context (Chinese: $F_{1,260} = 151.54$, $p < 0.001$; UK: $F_{1,210} = 137.80$, $p < 0.001$) were significant. Men allocated more mate dollars to physical attractiveness than women did and people allocated more mate dollars to physical attractiveness for short-term relationships than they did for long-term relationships (table 1). The interaction was not significant in either sample (Chinese: $F_{1,260} = 0.18$, $p = 0.671$; UK: $F_{1,210} = 1.73$, $p = 0.189$). These data support Prediction 1a, but not Prediction 1b.

## 6.2. Results of tests of Prediction 2

After data exclusions, 144 Chinese women, 151 Chinese men, 118 UK women and 120 UK men could be included in the final analyses of social status.

In both samples, the main effects of participant sex (Chinese: $F_{1,293} = 68.63$, $p < 0.001$; UK: $F_{1,236} = 12.01$, $p < 0.001$) and relationship context (Chinese: $F_{1,293} = 74.98$, $p < 0.001$; UK: $F_{1,236} = 31.11$, $p < 0.001$)

**Table 2.** Mean mate dollars allocated to social status by group (and s.d.).

| sample | relationship context | male participants | female participants |
|---|---|---|---|
| Chinese | long-term | 17.71 (8.49) | 27.19 (11.64) |
| Chinese | short-term | 12.85 (9.28) | 19.85 (12.16) |
| UK | long-term | 11.51 (8.66) | 14.65 (8.77) |
| UK | short-term | 8.37 (7.90) | 11.62 (7.83) |

**Table 3.** Mean mate dollars allocated to creativity by group (and s.d.).

| sample | relationship context | male participants | female participants |
|---|---|---|---|
| Chinese | long-term | 12.73 (6.59) | 14.44 (7.48) |
| Chinese | short-term | 9.83 (7.97) | 11.50 (8.52) |
| UK | long-term | 14.74 (7.21) | 14.97 (8.57) |
| UK | short-term | 11.34 (8.05) | 11.75 (8.38) |

**Table 4.** Mean mate dollars allocated to kindness by group (and s.d.).

| sample | relationship context | male participants | female participants |
|---|---|---|---|
| Chinese | long-term | 27.03 (10.90) | 25.37 (11.22) |
| Chinese | short-term | 20.35 (11.92) | 18.98 (11.23) |
| UK | long-term | 27.55 (9.74) | 31.46 (10.22) |
| UK | short-term | 18.78 (12.29) | 25.84 (11.45) |

were significant. Women allocated more mate dollars to social status than men did and people allocated more mate dollars to social status for long-term relationships than they did for short-term relationships (table 1). The interaction was not significant in either sample (Chinese: $F_{1,293} = 3.10$, $p = 0.080$; UK: $F_{1,236} = 0.01$, $p = 0.923$). These data support Prediction 2a, but not Prediction 2b.

## 6.3. Results of tests of Prediction 3

Consistent with Prediction 3, Chinese women allocated significantly more mate dollars to social status for long-term relationships than did UK women ($F_{1,260} = 93.52$, $p < 0.001$) (table 2).

# 7. Results of exploratory analyses

The same exclusion criteria detailed in the Data exclusions section were also applied to these exploratory analyses.

First, mate dollars allocated to creativity (table 3), kindness (table 4) and liveliness (table 5) were analysed in the same way as the tests for Predictions 1 and 2.

Analyses of creativity (table 3) showed significant effects of relationship context in both samples (Chinese: $F_{1,289} = 34.73$, $p < 0.001$; UK: $F_{1,235} = 42.55$, $p < 0.001$) and a significant effect of participant sex in the Chinese sample ($F_{1,289} = 5.08$, $p = 0.025$), but not the UK sample ($F_{1,235} = 0.12$, $p = 0.730$). The interaction was not significant in either sample (Chinese: $F_{1,289} = 0.00$, $p = 0.968$; UK: $F_{1,235} = 0.03$, $p = 0.852$). People showed stronger preferences for creativity in the long-term than short-term partners. Chinese women showed stronger preferences for creativity than did Chinese men.

**Table 5.** Mean mate dollars allocated to liveliness by group (and s.d.).

| sample | relationship context | male participants | female participants |
|---|---|---|---|
| Chinese | long-term | 17.12 (7.60) | 13.08 (7.64) |
| Chinese | short-term | 16.76 (8.23) | 14.55 (9.08) |
| UK | long-term | 19.53 (6.50) | 16.77 (7.11) |
| UK | short-term | 20.08 (9.25) | 17.09 (7.69) |

**Table 6.** Mean ideal partner age (adjusted for participant age).

| sample | relationship context | male participants | female participants |
|---|---|---|---|
| Chinese | long-term | 1.63 | 4.57 |
| Chinese | short-term | −0.28 | 1.84 |
| UK | long-term | 0.46 | 2.29 |
| UK | short-term | −0.30 | 1.64 |

Analyses of kindness (table 4) showed significant effects of relationship context in both samples (Chinese: $F_{1,307} = 88.48$, $p < 0.001$; UK: $F_{1,229} = 90.63$, $p < 0.001$) and a significant effect of participant sex in the UK sample ($F_{1,229} = 19.87$, $p < 0.001$), but not the Chinese sample ($F_{1,307} = 1.95$, $p = 0.164$). The interaction was significant in the UK sample ($F_{1,229} = 4.36$, $p = 0.038$), but not the Chinese sample ($F_{1,307} = 0.04$, $p = 0.837$). People showed stronger preferences for kindness in the long-term than short-term partners. UK women showed stronger preferences for kindness than did UK men.

Analyses of liveliness (table 5) showed significant effects of participant sex in both samples (Chinese: $F_{1,292} = 14.27$, $p < 0.001$; UK: $F_{1,175} = 8.62$, $p = 0.004$) and no significant effect of relationship context in either sample (Chinese: $F_{1,292} = 1.42$, $p = 0.234$; UK: $F_{1,175} = 0.49$, $p = 0.484$). The interaction was not significant in either sample (Chinese: $F_{1,292} = 3.79$, $p = 0.053$; UK: $F_{1,175} = 0.03$, $p = 0.858$). Men showed stronger preferences for liveliness than did women.

Next, the ideal partner age (adjusted for participant age by subtracting participant age from ideal age, i.e. larger values indicate a stronger preference for older partners) were analysed using ANOVA. In both samples, the main effects of participant sex (Chinese: $F_{1,310} = 83.94$, $p < 0.001$; UK: $F_{1,245} = 104.99$, $p < 0.001$) and relationship context (Chinese: $F_{1,310} = 165.14$, $p < 0.001$; UK: $F_{1,245} = 27.86$, $p < 0.001$) were significant. The interaction was significant in the Chinese sample ($F_{1,310} = 5.09$, $p = 0.020$), but not the UK sample ($F_{1,245} = 0.20$, $p = 0.660$). These results are summarized in table 6.

Finally, we tested how men and women valued attractiveness relative to the other traits on the mate dollars task. Data from the UK and Chinese samples were combined for these analyses and the results are reported in full in the supplemental materials. ANOVAs suggested that women valued physical attractiveness more than all other traits for short-term, but not long-term, relationships. This pattern occurred because women did not value physical attractiveness more than social status for long-term relationships. Men also valued physical attractiveness more than all other traits for short-term, but not long-term, relationships. This pattern occurred because men did not value physical attractiveness more than kindness for long-term relationships.

## 8. Discussion

We investigated the generality of previously reported effects of participant sex and relationship context on Chinese and UK participants' preferences for physical attractiveness and social status in potential mates. Confirmatory analyses supported our prediction (Prediction 1a) that men in both samples would show stronger preferences for physical attractiveness than women did and our prediction that women in both samples would show stronger preferences for social status than men did (Prediction 2a). These findings replicate sex differences in preferences for these traits that have been reported in previous research (e.g. [3,6,7]). By contrast, we found little evidence for the predictions (Predictions 1b and 2b) that the magnitude of these sex differences are moderated by the relationship

context for which partner preferences were expressed. These null results for the interactions between effects of participant sex and relationship context cannot easily be explained by a general failure of our relationship context manipulation, since our confirmatory analyses generally showed the same relatively strong effects of relationship context on preferences for physical attractiveness and social status that have been reported in previous research (reviewed in [3,4]). On the basis of these findings, we speculate that the interactions between participant sex and relationship context reported in some studies (e.g. [6]) are potentially not robust. Indeed, prominent theoretical perspectives do not straightforwardly predict such interactions [3,5].

Li *et al*. [6] previously reported that Singaporean women showed stronger preferences for social status in long-term partners than US women did. A confirmatory analysis of women's preferences for social status replicated this pattern in a comparison of Chinese and UK women's partner preferences, supporting the suggestion that this pattern represents a general difference in the extent to which women in Eastern and Western countries value social status in long-term partners [6].

Exploratory analyses suggested that, for both the Chinese and UK samples, women had stronger preferences for older partners than men did and that people had stronger preferences for older partners for long-term relationships than short-term relationships. These replicate the results of previous studies (reviewed in [4,5]). In other exploratory analyses, we examined how men and women prioritized attractiveness relative to other traits for long- and short-term relationships. Both men and women valued physical attractiveness more than the other traits on the mate dollars task for short-term relationships, but not long-term relationships. Women did not differ significantly in their preferences for social status and physical attractiveness for long-term relationships and men did not differ significantly in their preferences for kindness and physical attractiveness for long-term relationships. The results of these exploratory analyses should be treated cautiously, however, since many of the effects would not survive correction for multiple comparisons and may then be false positives.

Our exploratory analyses of age preferences showed that women had stronger preferences for older mates than did men. This replicates a well-established pattern of results in the mate preferences literature [3,5]. However, the men in our study did (on average) express a preference for mates older than themselves, particularly for long-term relationships. This is a surprising result, since men typically prefer mates younger than themselves [3,5]. Whether or not this is a pattern that replicates in similar samples (e.g. university students) is a question for future research.

In conclusion, our confirmatory analyses present further evidence that sex differences in preferences for physical attractiveness and social status in potential mates occur in a wide range of cultures. This is consistent with the suggestion that they at least partly reflect biological universals, such as sex differences in age-related decline in fertility and costs of pregnancy and lactation [1,3,6]. However, the difference in the extent to which Chinese and UK women valued social status in potential mates suggests that factors other than biological universals also influence mate preferences.

Ethics. All procedures were approved by the University of Glasgow, School of Psychology, Ethics Committee and all participants provided informed consent.

Data accessibility. Data and analysis code are archived at https://osf.io/rkstx/.

Authors' contributions. All authors contributed to designing the study. L.Z. and H.W. collected the data. L.Z., A.J.L. and L.M.D. wrote an analysis code and carried out statistical analyses. L.Z. and B.C.J. wrote the first draft of the manuscript. L.M.D. designed testing interfaces. All authors revised and approved the manuscript.

Competing interests. We declare we have no competing interests.

Funding. This research was funded by an ERC grant (KINSHIP) awarded to L.M.D.

Acknowledgement. The authors thank Julia Stern and Alex Jones for constructive comments on previous versions of this manuscript.

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
