## [Reviewer comments · Royal Society Open Science]

Review History

RSOS-181000.R0 (Original submission)

Decision letter (RSOS-181000.R0)

04-Jul-2018

Dear Dr Jones,

I write you in regards to manuscript RSOS-181000 entitled "Chinese and UK participants' preferences for physical attractiveness and social status in potential mates (Stage 1 Registered Report)" which you submitted to Royal Society Open Science.

We routinely triage submissions for scientific soundness, clarity and general adherence to the Registered Reports guidelines. For submissions that have promise but are not yet suitable for in-depth Stage 1 review, we offer feedback to help authors maximise the chances that reviewers will respond positively to a resubmission.

We have concluded that your submission is not yet suitable for in-depth review and has therefore been rejected at this time, but we believe it will be suitable once several issues are addressed. We therefore invite a resubmission. Further comments from the Associate Editor may be found at the end of this letter.

If you wish to revise your manuscript in light of the below comments please submit your manuscript as a new submission and mention this previous manuscript ID in your covering letter. You should also provide a detailed response to the below comments in the cover letter.

Please note that Royal Society Open Science will introduce article processing charges for all new submissions received from 1 January 2018. Registered Reports submitted and accepted after this date will ONLY be subject to a charge if they subsequently progress to and are accepted as Stage 2 Registered Reports. If your manuscript is submitted and accepted for publication after 1 January 2018 (i.e. as a full Stage 2 Registered Report), you will be asked to pay the article processing charge, unless you request a waiver and this is approved by Royal Society Publishing. You can find out more about the charges at <http://rsos.royalsocietypublishing.org/page/charges>. Should you have any queries, please contact openscience@royalsociety.org.

Thank you for considering Royal Society Open Science for the publication of your registered report.

Kind regards,
Thadcha Retneswaran
Editorial Coordinator
Royal Society Open Science
openscience@royalsociety.org

on behalf of Chris Chambers (Registered Reports Editor, Royal Society Open Science)
openscience@royalsociety.org

Comments to the Author:

Should you decide to resubmit, please address the following issues:

1. Power analysis/sampling plan. Where authors propose frequentist analytic methods, all hypothesis tests should be associated with a power analysis to detect either (a) the smallest effect of theoretical interest, or (b) the lower bound estimate of the expected effect size from a representative sample of the relevant prior literature. This condition is not yet met in your proposal. As an alternative to frequentist testing, you may wish to consider alternative Bayesian inferential methods that permit optional stopping to maximise the efficiency of sampling and to also permit positive conclusions about negative results. For further information, please see Dienes 2014 <http://journal.frontiersin.org/article/10.3389/fpsyg.2014.00781/full>

2. Please ensure that there is a direct correspondence between the proposed hypotheses, the power analyses (or alternative sampling plans), and the proposed statistical tests. At present this connection is not sufficiently explicit to proceed to in-depth review. The power analysis needs to map directly on to the proposed statistical tests, each of which need to correspond to one of the specified hypotheses. For maximum clarity, we recommend numbering the hypotheses at the end of the Introduction (e.g. in a list, as you have done) and then similarly listing in the analysis plan which statistical test addresses which specific hypothesis, using which dependent variables, and reporting how each test or test component achieves the designated power level (we recommend

achieving a minimum of 90% but a lower level can be proposed; where including power analysis please be sure to include and justify all input parameters, such as for ANOVA the assumed correlation between repeated measures where applicable, and the method used for estimation).

3. Please ensure that all procedures are described in sufficient detail to enable direct replication without the reader needing to review prior work. For instance, how are the various response categories – attractiveness, creativity etc – presented to participants? Perhaps provide a visual schematic of the procedure, details of translations etc. Registered Reports must be internally reproducible the maximum reasonable extent.
4. One of the key criteria that reviewers are asked to assess in Stage 1 RRs is "Whether the authors have considered sufficient outcome-neutral conditions (e.g. absence of floor or ceiling effects; positive controls; other quality checks) for ensuring that the results obtained are able to test the stated hypotheses.", and successfully passing such tests is an editorial criterion at Stage 2 following completion of the study. Your protocol does not seem to propose any such tests - therefore please consider whether such positive controls or data quality checks are necessary (they might not be for some designs), and if possible how they might be included in the design. Such tests should be orthogonal of the main hypotheses and, where they involve a hypothesis test, should be accompanied by a power analysis or alternative (e.g. Bayesian) sampling plan. We recommend outlining any such tests in a separate section.
5. Please ensure that exclusion criteria for data within and across participants are comprehensively pre-specified as it usually not possible to adjust these for pre-registered analyses after provisional acceptance is granted. For example, you note that "Responses more than three standard deviations from the mean for the sample will be excluded from the dataset prior to analyses" (p5). What does "the sample" refer to? The total sample? Each national sample separately? For which DVs or averaged across which DVs? Within which conditions or sub-conditions?
6. Please provide details of ethical approval in the main text of the manuscript.
7. Does recruitment involve any screening for age, sexual orientation, or ethnic status within each country? Please explain in the manuscript.
8. Given that all hypotheses relate to social status and physical attractiveness, what is the rationale for including creativity, kindness, and liveliness as measures? Are these for exploratory analysis only?
9. In general, to what extent might cultural differences in outcomes be explained by slightly different definitions or conceptualisations of the terms between languages? Do you have confirmation, for instance, that complex concepts like "social status" are understood and conceptualised identically between countries?

We hope this feedback is helpful and look forward to receiving your revised submission.

Author's Response to Decision Letter for (RSOS-181000.R0)

See Appendix A.

RSOS-181243.R0

Review form: Reviewer 1

Is the language acceptable?

Yes

Do you have any ethical concerns with this paper?

No

Have you any concerns about statistical analyses in this paper?

No

Recommendation?

Major revision

Comments to the Author(s)

This brief manuscript consists of a first-stage registered report, proposing to utilize a budget allocation methodology to cross-culturally investigate mate preferences. The manuscript reads well, the proposed study poses a potentially interesting examination, and the projected power seems to be adequate. Nonetheless, there are some areas that can benefit from further attention in a revision.

Although I like brief manuscripts, I feel that the authors may have erred on the side of underselling the current framework and contribution. For the framework, what are the alleged methodological limitations and how does the budget allocation methodology address them? Most mate preference studies ask participants to rate or rank the importance of various traits. In those cases, participants can rate all traits highly or they can rate or rank them after assuming (understandably) that certain traits are already at high levels and thus, are not important, thereby potentially obscuring differences between traits and between sexes. For instance, most college students have an average or higher level of social status, so this trait may not seem to be particularly important when looking for mates.

The budget allocation method was designed to address this major shortcoming (in conjunction with testing hypotheses derived from a biological perspective, which was not adequately tested or supported using simple surveys) by using a budget to constrain overall choice (such that only a low level of each trait can be obtained, on average), thereby forcing participants to reveal what they prioritize the most among a set of choices. The earliest paper in the budget-allocation series (Li et al., 2002) explains this background and reasoning thoroughly, in terms of necessities versus luxuries. The authors should explain a bit of this in the current manuscript (including that the DVs here represent not just preferences but priorities) to more fully convey the theoretical importance of the chosen framework.

Also, although an extension of the budget allocation methodology to investigate mate preferences in other countries might be a reasonable undertaking, the UK and China samples seem, at least on the surface, to be rather similar to the countries sampled in the Li et al. (2011) paper - the US and Singapore. It would help substantiate the manuscript's importance/contribution if the authors can make clear how the currently selected two countries will add value to what has already been shown. Although it might be worth simply adding additional countries to further substantiate a Western versus Eastern comparison (citations for when other researchers have done that for a similar question or other social psychological topic would help), there might be

additional (ideally, theory-based) reasons for including these countries that the authors can consider and explain.

In the Li et al. (2011) paper (and, I believe, papers that preceded that one where the methodology was originally introduced), the studies informed participants that the mate dollars allowed them to purchase percentile points on each trait dimension. That is, traits start out at the 0th percentile, and spending, say, 40 mate dollars on a trait means that you've purchased 40 percentile points worth of that trait (i.e., a potential mate who is above 40% of the population on that trait). This information is further explained in materials on the first author's website and should be included in the current study, as it indicates to participants what the mate dollars signify.

Although the authors are primarily investigating sex differences, they also have an opportunity to test for relative trait preferences by sex, which are also an important part of the mate preferences literature. That is, they can also investigate (and make predictions about) what specific traits are preferred the most in each context, and whether such relative preferences differ by sex. As found in Li et al. (2011) and other studies, men may value physical attractiveness more than other traits for both short- and long-term contexts, and women may value physical attractiveness more than other traits in short-term, but not long-term, contexts. These predictions indicate that, although (as shown by the authors' current hypotheses) men may more greatly value physical attractiveness than women do for short-term mates, both sexes value physical attractiveness the most in short-term contexts (though purportedly for different reasons – Li & Kenrick, 2006).

Review form: Reviewer 2

Is the language acceptable?

Yes

Do you have any ethical concerns with this paper?

No

Have you any concerns about statistical analyses in this paper?

No

Recommendation?

Major revision

Comments to the Author(s)

The proposed study would represent a nice replication of Li et al's study "Mate preferences in the US and Singapore: A cross-cultural test of the mate preference priority model" (2011). My main concerns are about the logic and theoretical background of two of the predictions (1b and 3). I also think the methods could be improved to ensure that UK and Chinese participants are rating the same concepts, a key point for this kind of cross-cultural studies.

One of the authors' hypothesis is that "Men are hypothesized to show stronger preferences for physical attractiveness in potential mates than women are, particularly when assessing the attractiveness of potential mates for short-term relationships." I agree with the first part of the hypothesis, but not with the second. Indeed, as the authors explain in the first paragraph of the introduction "When assessing men's attractiveness as short-term partners, however, resources are thought to be less important and women are hypothesized to prefer men displaying cues that

they are in good physical condition and will father healthy children". This sentence seems to be in contradiction with the authors' hypothesis: for short-term male partners, physical attractiveness becomes more important to women (as a good physical condition is linked to physical attractiveness). Thus, in the case of short-term relationships, women's preferences are becoming MORE like men's. Or, perhaps in the short-term context the preference for physical attractiveness is also increasing for men, such that the difference between men and women might stay the same. Note that this is in accordance with Li et al's prediction: "For short-term mates, both sexes have been hypothesized to place high value on physical attractiveness".

In the end, I think the authors should revise their prediction 1b. I am aware that Li et al. previously found a larger sex difference for short-term partners, but if this is the only reason for the prediction 1b, the authors should clearly state it, adding that it is in conflict with the theory. Otherwise, the authors need to explain why they are particularly expecting a larger difference between male and female preferences for physical attractiveness in the case of short-term partners compared to long-term partners.

I do not understand the theory behind prediction 3: "Chinese women will allocate significantly more mate dollars to social status than UK women will." To support this prediction, the authors state that "Li et al. [5] suggested this latter result was consistent with social status being more important for social interactions generally in Eastern than Western cultures [6]". This does not explain why this is only found for women. Moreover, it would be nice to give more theoretical background on why social status is more important in Eastern than Western cultures.

I do like that the authors are using the Psychological Science Accelerator's translation procedure. However, in this particular case, I would add another step. Step 4, where members of the fluent group are asked how they perceive and understand the translation, should also be done for the English version. More precisely, I would ask two groups of native speakers of each language to explain the different concepts and to give synonyms for each word. Then, the answers for the two groups would be compared. Moreover, each group should include more than only two individuals (as currently proposed). This additional step requires a little more work, but I think it is necessary for a study based on a cross-cultural comparison. Without a very solid check of the understanding of the concepts through a sufficient sample, any difference between the two country has a large likelihood of simply being a false positive.

To have an even more solid design, I would also add some manipulation checks at the end of the questionnaires. The authors could add a list of synonyms and related words for each concept (this list could derive from the discussions during the translation procedure). Each participant would choose a certain number of words which illustrate best the concepts, and then the answers would be compared. Note that, if it appears that the two populations have slightly different interpretations of the concepts, this would still be an interesting result. It could potentially give more light on previous studies showing cross-cultural differences on preferences.

Participant recruitment: It is not clear how the authors will define UK or Chinese participants. Are they going to ask for individuals' ethnicity? What would be the exact question do define it?

The authors explain that "because fertility declines faster with age in women than men, men are hypothesized to show stronger preferences for cues of reproductive capacity (e.g., youth) in women than women do when assessing the attractiveness of potential mates".

Indeed, age is a very important factor (maybe the most important) influencing women's reproductive capacity (and thus physical attractiveness). But this is not the only one. The authors could add that health, nutritional status and previous reproductive events also influence fertility in a stronger way for women than men, because of the large physiological costs of pregnancy and lactation. All these factors influence physical appearance and, consequently, women's physical attractiveness.

“Second, because women bear greater costs of obligatory parental investment (i.e., pregnancy and lactation) than men do [2], women are hypothesized to show stronger preferences for cues of capability to invest resources in offspring when assessing men’s attractiveness as long-term partners [1,3].”

The link between the two parts (costs of pregnancy/lactation and preferences for capability to invest resources) is not direct. I guess the missing link is that men’s resources can help to compensate the costs of pregnancy and lactation (direct benefits). More precisely, there are two different aspects that make resource acquisition crucial: pregnant and lactating women have increased needs (in calories, protection, etc.), AND their abilities to access these resources are diminished compared to non-pregnant or non-lactating women. I am sure this is what the authors had in mind, but I think it is important to be precise in the theoretical background.

At the end of the document, the authors explain that the traits “creativity”, “kindness”, and “liveliness” are only included in the study because they were included in Li et al. I think they should make this point before, for example at the beginning of the methods section.

Review form: Reviewer 3

Is the language acceptable?

Yes

Do you have any ethical concerns with this paper?

No

Have you any concerns about statistical analyses in this paper?

No

Recommendation?

Accept with minor revision

Comments to the Author(s)

This registered report deals with the question whether cross cultural (sex) differences in mate preferences are robust. The planned study is a replication of Li and colleagues (2011; PAID). The manuscript follows a clear line that provides necessary information for a registered report. I especially liked the analysis plan and the code that was provided by the authors. The stated hypotheses are plausible. I think this study will make a good contribution to the literature. Below I outline rather minor points that might improve the manuscript.

1. The original study by Li et al. (2011) compared a sample from the US with a sample from Singapore. Authors should address potential limitations that come with a replication that uses different samples (not all western cultures are the same, just as not all eastern cultures are the same).
2. I think that Li et al. have developed/ used the budget allocation task before 2011 (p. 3 line 22), I think it was in their 2002 (JPSP) paper.
3. I think authors should state why the budget-allocation task is assumed to be less affected by methodological limitations compared to e.g. rating tasks when comparing different cultures (p. 3 lines 13-23).
4. Authors state that they will only use WHITE participants from the UK to test their predictions (p.7 line 14; if I understood correctly). Why? Will they exclude all other participants? What about the Chinese participants? If they limit one sample they should probably also limit the other

sample as well and state this as an exclusion criteria. However, in the original Li et al. study, participants of different ethnicities were included.

5. Regarding prediction 3: Please specify if you predict this for short-term or for long-term mate preferences. If I remember it correctly, Li et al. (2011) did only find a significant difference for long-term, but not for short-term preferences (although the effect was –descriptively– in the same direction for short-term preferences).

6. The planned sample size is about the same size as the sample of the original study, however, it differs a bit regarding the amount of women/men sampled. Li et al. had 124 UK women (83 men) and 126 women (74 men) from Singapore. The authors should think about recruiting more participants in total, but at least the same amount of women that the original sample size had to make sure that they achieve the same power to replicate the effects for predictions 2 and 3 (I think 125 women and men from each country would be nice – but the more the better). Given that the task doesn't seem to require a huge amount of time and money, I think increasing the sample size might be possible. If not, please explain why and state it as a limitation.

7. Please describe any other assessed variables (like demographics) in the “procedure” part. E.g. will you assess participant's age or relationship status?

8. It would be helpful to compare the results from the power analyses to the previous literature. Is the effect size in the range of previously reported effect sizes? Is it reasonable to assume that the effect will be around that size?

Review form: Reviewer 4

Is the language acceptable?

Yes

Do you have any ethical concerns with this paper?

No

Have you any concerns about statistical analyses in this paper?

No

Recommendation?

Major revision

Comments to the Author(s)

Although I think this is terrific to conduct this study, I have problems with both the theoretical rationale as well as with the instruments. The background literature review and its conclusion are also problematic.

First, the predicted sex differences in long-term mating contexts (e.g., marriage, committed mating) are clear. There is no sense, however, in which the empirical evidence for these sex differences is ‘mixed,’ as the authors claim. This set of sex differences – in importance of attractiveness, youth, and financial resources – is among the most replicable sex differences in the entire field of psychology. It started with Buss's (1989) study of 37 cultures, but has been replicated more than a hundred times in an astonishing variety of other cultures. The authors may wish to check some review articles or books to obtain the many times these sex differences have been replicated. I've reviewed many of them in my *Evolutionary Psychology* (2015)

textbook, as well as in an ‘in press’ chapter in the *Annual Review of Psychology* (“Mate Preferences and Their Behavioral Manifestations” Buss & Schmitt, in press). I would be happy to provide the latter to the authors if they are interested.

The many replications of these sex differences include the two countries proposed for the current study – China and the UK. In fact, the sex differences in mate preferences first discovered in China and the UK in 1989 have recently been replicated in China using an extremely large sample size--(Chang, L., Wang, Y., Shackelford, T. K., & Buss, D. M. (2011). Chinese mate preferences: Cultural evolution and continuity across a quarter of a century. *Personality and Individual Differences*, 50(5), 678-683.) Moreover, this 2011 study of mate preferences in China had a sample size (N = 1,060 in 2008 and N = 600 in the mi-1980’s sample) more than 5 times that proposed in the current study.

Second, the theoretical rationale for predictions about sex differences in short-term mating is problematic. There have been several competing hypotheses proposed for the adaptive function of women’s short-term (ST) mating. These include: (1) good genes (for which attractiveness has been proposed as one marker), (2) mate switching, (3) immediate resources extraction (Buss & Schmitt, 1993, a key paper the author do not cite), (4) access to a higher status pool of potential mates, etc. The predictions for women’s ST mate preferences depend critically upon which hypothesis is being tested for the adaptive function. If it is #1 [which the authors seem to endorse], advocates of that hypothesis predict that women should dramatically elevate their mate preferences for physical attractiveness, which of course would diminish or attenuate any sex differences in mate preference in ST mating contexts [contrary to the author’s prediction]. If the hypothesis is #4, then women might elevate the importance of status in a potential mate. The key point is that the author’s predictions about sex differences in ST mating contexts are problematic and not well grounded theoretically. They really need to be clarified for coherent predictions to be made.

It should be noted that, empirically, it is the case that (in delimited cultures thus far studied), women do elevate the importance they attach to physical attractiveness in ST mating contexts (see Buss & Schmitt, 1993 for the first documentation of this finding).

Third, the use of ‘social status’ as the sole operationalization is problematic. Although some others have used this, the ‘status’ variable is highly problematic, in part because it is used in different senses in different cultures, and is inherently a vague construct unless is defined explicitly. Apropos this point, Buss (1989) found universal or near universal sex differences in the importance attached to ‘good financial prospects,’ but not universal sex differences in the importance attached to ‘status’ [although most cultures did show a sex difference in the predicted direction]. Because, as the author’s state, ‘financial resources’ is much closer to the theoretical variable of interest than ‘status,’ I recommend that the authors use ‘good financial resources’ as the key operationalization of that variable rather than status. That would be a much cleaner test of the sex differences prediction.

Fourth, although the budget allocation method has some advantages over rating scales, it also has disadvantages. By being a forced choice procedure, it preclude discovering many patterns of results that might be present (e.g., if people valued two qualities highly, and placed no value on the other qualities). That is the reason that Buss (1989) used two different methods – a rating method AND a ranking method, which is structurally nearly identical to the ‘budget allocation’ method in that it forces participants to make tradeoffs. Both (budget allocation and ranking) are forced choice methods and have the same strengths and limitations.

Fifth, given the modest scope of the proposed study, I see no reason why the authors could not include a measure of age preference in a mate, or age differences preferred between self and

mate. This would increase the value of the study in at least two ways. First, it would provide a critical test of a key hypothesis about sex differences in mate preferences – with men predicted to prefer younger partners than do women. Second, given the LT v. ST design, it would allow a test of the prediction that men shift their age preferences to be somewhat younger in LT mating contexts (i.e., women of higher reproductive value or future reproductive potential, perhaps late teens) and slightly higher in ST mating contexts (i.e., closer to peak fertility, which would be mid-twenties). So far, this latter hypothesis has not been sufficiently tested to determine whether it has any empirical support. This would add greatly to the novelty of the current proposed research.

These are the main suggestions. However, I'll add a minor one. It is my understanding that one of the authors has criticized mate preference research as being too heavily focused on heterosexual populations. Thus, I was surprised that the current study did not include a sample of non-heterosexual participants. Perhaps the proposed N in the current studies is too modest to include large enough samples of the latter, but given the relatively brief procedures proposed, I don't see any reason why the sample size could not be expanded.

In short, I think the study design and theoretical rationale really need to be improved in order for the study to make a novel and high quality contribution to the large extant scientific literature on human mate preferences.

Decision letter (RSOS-181243.R0)

30-Aug-2018

Dear Dr Jones,

The Editors assigned to your stage one Registered Report ("Chinese and UK participants' preferences for physical attractiveness and social status in potential mates (Stage 1 Registered Report)") have now received comments from reviewers. We would like you to revise your paper in accordance with the referee and editors suggestions which can be found below (not including confidential reports to the Editor). Please note this decision does not guarantee eventual acceptance.

Please note that Royal Society Open Science charge article processing charges for all new submissions that are accepted for publication. Charges will also apply to papers transferred to Royal Society Open Science from other Royal Society Publishing journals, as well as papers submitted as part of our collaboration with the Royal Society of Chemistry (<http://rsos.royalsocietypublishing.org/chemistry>). If your manuscript is newly submitted and

subsequently accepted for publication, you will be asked to pay the article processing charge, unless you request a waiver and this is approved by Royal Society Publishing. You can find out more about the charges at <http://rsos.royalsocietypublishing.org/page/charges>. Should you have any queries, please contact openscience@royalsociety.org.

on behalf of Professor Chris Chambers (Registered Reports Editor, Royal Society Open Science)
openscience@royalsociety.org

Associate Editor Comments to Author (Professor Chris Chambers):

Associate Editor: 1

Comments to the Author:

This letter follows from my informal email of last week in which I relayed the editorial decision (Scholar One have now repaired the RSOS manuscript handling system). Four expert reviewers have now provided in-depth assessments. The reviews are mostly positive but also quite critical, raising issues that cut across the full range of Stage 1 criteria, from the rationale for the supporting theory and hypotheses, to the diversity of the recruited sample, to the clarity of the procedures and analyses. From an editorial perspective, all of the issues raised appear readily addressable through revision or rebuttal. A Major Revision is therefore invited.

Comments to Author:

Reviewer: 1

Comments to the Author(s)

This brief manuscript consists of a first-stage registered report, proposing to utilize a budget allocation methodology to cross-culturally investigate mate preferences. The manuscript reads well, the proposed study poses a potentially interesting examination, and the projected power seems to be adequate. Nonetheless, there are some areas that can benefit from further attention in a revision.

Although I like brief manuscripts, I feel that the authors may have erred on the side of underselling the current framework and contribution. For the framework, what are the alleged methodological limitations and how does the budget allocation methodology address them? Most mate preference studies ask participants to rate or rank the importance of various traits. In those cases, participants can rate all traits highly or they can rate or rank them after assuming (understandably) that certain traits are already at high levels and thus, are not important, thereby potentially obscuring differences between traits and between sexes. For instance, most college students have an average or higher level of social status, so this trait may not seem to be particularly important when looking for mates.

The budget allocation method was designed to address this major shortcoming (in conjunction with testing hypotheses derived from a biological perspective, which was not adequately tested or supported using simple surveys) by using a budget to constrain overall choice (such that only

a low level of each trait can be obtained, on average), thereby forcing participants to reveal what they prioritize the most among a set of choices. The earliest paper in the budget-allocation series (Li et al., 2002) explains this background and reasoning thoroughly, in terms of necessities versus luxuries. The authors should explain a bit of this in the current manuscript (including that the DVs here represent not just preferences but priorities) to more fully convey the theoretical importance of the chosen framework.

Also, although an extension of the budget allocation methodology to investigate mate preferences in other countries might be a reasonable undertaking, the UK and China samples seem, at least on the surface, to be rather similar to the countries sampled in the Li et al. (2011) paper – the US and Singapore. It would help substantiate the manuscript's importance/contribution if the authors can make clear how the currently selected two countries will add value to what has already been shown. Although it might be worth simply adding additional countries to further substantiate a Western versus Eastern comparison (citations for when other researchers have done that for a similar question or other social psychological topic would help), there might be additional (ideally, theory-based) reasons for including these countries that the authors can consider and explain.

In the Li et al. (2011) paper (and, I believe, papers that preceded that one where the methodology was originally introduced), the studies informed participants that the mate dollars allowed them to purchase percentile points on each trait dimension. That is, traits start out at the 0th percentile, and spending, say, 40 mate dollars on a trait means that you've purchased 40 percentile points worth of that trait (i.e., a potential mate who is above 40% of the population on that trait). This information is further explained in materials on the first author's website and should be included in the current study, as it indicates to participants what the mate dollars signify.

Although the authors are primarily investigating sex differences, they also have an opportunity to test for relative trait preferences by sex, which are also an important part of the mate preferences literature. That is, they can also investigate (and make predictions about) what specific traits are preferred the most in each context, and whether such relative preferences differ by sex. As found in Li et al. (2011) and other studies, men may value physical attractiveness more than other traits for both short- and long-term contexts, and women may value physical attractiveness more than other traits in short-term, but not long-term, contexts. These predictions indicate that, although (as shown by the authors' current hypotheses) men may more greatly value physical attractiveness than women do for short-term mates, both sexes value physical attractiveness the most in short-term contexts (though purportedly for different reasons – Li & Kenrick, 2006).

Reviewer: 2

Comments to the Author(s)

The proposed study would represent a nice replication of Li et al's study "Mate preferences in the US and Singapore: A cross-cultural test of the mate preference priority model" (2011). My main concerns are about the logic and theoretical background of two of the predictions (1b and 3). I also think the methods could be improved to ensure that UK and Chinese participants are rating the same concepts, a key point for this kind of cross-cultural studies.

One of the authors' hypothesis is that "Men are hypothesized to show stronger preferences for physical attractiveness in potential mates than women are, particularly when assessing the attractiveness of potential mates for short-term relationships." I agree with the first part of the hypothesis, but not with the second. Indeed, as the authors explain in the first paragraph of the introduction "When assessing men's attractiveness as short-term partners, however, resources

are thought to be less important and women are hypothesized to prefer men displaying cues that they are in good physical condition and will father healthy children". This sentence seems to be in contradiction with the authors' hypothesis: for short-term male partners, physical attractiveness becomes more important to women (as a good physical condition is linked to physical attractiveness). Thus, in the case of short-term relationships, women's preferences are becoming MORE like men's. Or, perhaps in the short-term context the preference for physical attractiveness is also increasing for men, such that the difference between men and women might stay the same. Note that this is in accordance with Li et al's prediction: "For short-term mates, both sexes have been hypothesized to place high value on physical attractiveness".

In the end, I think the authors should revise their prediction 1b. I am aware that Li et al. previously found a larger sex difference for short-term partners, but if this is the only reason for the prediction 1b, the authors should clearly state it, adding that it is in conflict with the theory. Otherwise, the authors need to explain why they are particularly expecting a larger difference between male and female preferences for physical attractiveness in the case of short-term partners compared to long-term partners.

I do not understand the theory behind prediction 3: "Chinese women will allocate significantly more mate dollars to social status than UK women will." To support this prediction, the authors state that "Li et al. [5] suggested this latter result was consistent with social status being more important for social interactions generally in Eastern than Western cultures [6]". This does not explain why this is only found for women. Moreover, it would be nice to give more theoretical background on why social status is more important in Eastern than Western cultures.

I do like that the authors are using the Psychological Science Accelerator's translation procedure. However, in this particular case, I would add another step. Step 4, where members of the fluent group are asked how they perceive and understand the translation, should also be done for the English version. More precisely, I would ask two groups of native speakers of each language to explain the different concepts and to give synonyms for each word. Then, the answers for the two groups would be compared. Moreover, each group should include more than only two individuals (as currently proposed). This additional step requires a little more work, but I think it is necessary for a study based on a cross-cultural comparison. Without a very solid check of the understanding of the concepts through a sufficient sample, any difference between the two country has a large likelihood of simply being a false positive.

To have an even more solid design, I would also add some manipulation checks at the end of the questionnaires. The authors could add a list of synonyms and related words for each concept (this list could derive from the discussions during the translation procedure). Each participant would choose a certain number of words which illustrate best the concepts, and then the answers would be compared. Note that, if it appears that the two populations have slightly different interpretations of the concepts, this would still be an interesting result. It could potentially give more light on previous studies showing cross-cultural differences on preferences.

Participant recruitment: It is not clear how the authors will define UK or Chinese participants. Are they going to ask for individuals' ethnicity? What would be the exact question do define it?

The authors explain that "because fertility declines faster with age in women than men, men are hypothesized to show stronger preferences for cues of reproductive capacity (e.g., youth) in women than women do when assessing the attractiveness of potential mates".

Indeed, age is a very important factor (maybe the most important) influencing women's reproductive capacity (and thus physical attractiveness). But this is not the only one. The authors could add that health, nutritional status and previous reproductive events also influence fertility in a stronger way for women than men, because of the large physiological costs of pregnancy and lactation. All these factors influence physical appearance and, consequently, women's physical attractiveness.

“Second, because women bear greater costs of obligatory parental investment (i.e., pregnancy and lactation) than men do [2], women are hypothesized to show stronger preferences for cues of capability to invest resources in offspring when assessing men’s attractiveness as long-term partners [1,3].”

The link between the two parts (costs of pregnancy/lactation and preferences for capability to invest resources) is not direct. I guess the missing link is that men’s resources can help to compensate the costs of pregnancy and lactation (direct benefits). More precisely, there are two different aspects that make resource acquisition crucial: pregnant and lactating women have increased needs (in calories, protection, etc.), AND their abilities to access these resources are diminished compared to non-pregnant or non-lactating women. I am sure this is what the authors had in mind, but I think it is important to be precise in the theoretical background.

At the end of the document, the authors explain that the traits “creativity”, “kindness”, and “liveliness” are only included in the study because they were included in Li et al. I think they should make this point before, for example at the beginning of the methods section.

Reviewer: 3

Comments to the Author(s)

This registered report deals with the question whether cross cultural (sex) differences in mate preferences are robust. The planned study is a replication of Li and colleagues (2011; PAID). The manuscript follows a clear line that provides necessary information for a registered report. I especially liked the analysis plan and the code that was provided by the authors. The stated hypotheses are plausible. I think this study will make a good contribution to the literature. Below I outline rather minor points that might improve the manuscript.

1. The original study by Li et al. (2011) compared a sample from the US with a sample from Singapore. Authors should address potential limitations that come with a replication that uses different samples (not all western cultures are the same, just as not all eastern cultures are the same).
2. I think that Li et al. have developed/ used the budget allocation task before 2011 (p. 3 line 22), I think it was in their 2002 (JPSP) paper.
3. I think authors should state why the budget-allocation task is assumed to be less affected by methodological limitations compared to e.g. rating tasks when comparing different cultures (p. 3 lines 13-23).
4. Authors state that they will only use WHITE participants from the UK to test their predictions (p.7 line 14; if I understood correctly). Why? Will they exclude all other participants? What about the Chinese participants? If they limit one sample they should probably also limit the other sample as well and state this as an exclusion criteria. However, in the original Li et al. study, participants of different ethnicities were included.
5. Regarding prediction 3: Please specify if you predict this for short-term or for long-term mate preferences. If I remember it correctly, Li et al. (2011) did only find a significant difference for long-term, but not for short-term preferences (although the effect was –descriptively- in the same direction for short-term preferences).
6. The planned sample size is about the same size as the sample of the original study, however, it differs a bit regarding the amount of women/men sampled. Li et al. had 124 UK women (83 men) and 126 women (74 men) from Singapore. The authors should think about recruiting more participants in total, but at least the same amount of women that the original sample size had to make sure that they achieve the same power to replicate the effects for predictions 2 and 3 (I think 125 women and men from each country would be nice – but the more the better). Given that the task doesn’t seem to require a huge amount of time and money, I think increasing the sample size might be possible. If not, please explain why and state it as a limitation.

7. Please describe any other assessed variables (like demographics) in the “procedure” part. E.g. will you assess participant’s age or relationship status?

8. It would be helpful to compare the results from the power analyses to the previous literature. Is the effect size in the range of previously reported effect sizes? Is it reasonable to assume that the effect will be around that size?

Reviewer: 4

Comments to the Author(s)

Although I think this is terrific to conduct this study, I have problems with both the theoretical rationale as well as with the instruments. The background literature review and its conclusion are also problematic.

First, the predicted sex differences in long-term mating contexts (e.g., marriage, committed mating) are clear. There is no sense, however, in which the empirical evidence for these sex differences is ‘mixed,’ as the authors claim. This set of sex differences – in importance of attractiveness, youth, and financial resources – is among the most replicable sex differences in the entire field of psychology. It started with Buss’s (1989) study of 37 cultures, but has been replicated more than a hundred times in an astonishing variety of other cultures. The authors may wish to check some review articles or books to obtain the many times these sex differences have been replicated. I’ve reviewed many of them in my *Evolutionary Psychology* (2015) textbook, as well as in an ‘in press’ chapter in the *Annual Review of Psychology* (“Mate Preferences and Their Behavioral Manifestations” Buss & Schmitt, in press). I would be happy to provide the latter to the authors if they are interested.

The many replications of these sex differences include the two countries proposed for the current study – China and the UK. In fact, the sex differences in mate preferences first discovered in China and the UK in 1989 have recently been replicated in China using an extremely large sample size--(Chang, L., Wang, Y., Shackelford, T. K., & Buss, D. M. (2011). Chinese mate preferences: Cultural evolution and continuity across a quarter of a century. *Personality and Individual Differences*, 50(5), 678-683.) Moreover, this 2011 study of mate preferences in China had a sample size (N = 1,060 in 2008 and N = 600 in the mi-1980’s sample) more than 5 times that proposed in the current study.

Second, the theoretical rationale for predictions about sex differences in short-term mating is problematic. There have been several competing hypotheses proposed for the adaptive function of women’s short-term (ST) mating. These include: (1) good genes (for which attractiveness has been proposed as one marker), (2) mate switching, (3) immediate resources extraction (Buss & Schmitt, 1993, a key paper the author do not cite), (4) access to a higher status pool of potential mates, etc. The predictions for women’s ST mate preferences depend critically upon which hypothesis is being tested for the adaptive function. If it is #1 [which the authors seem to endorse], advocates of that hypothesis predict that women should dramatically elevate their mate preferences for physical attractiveness, which of course would diminish or attenuate any sex differences in mate preference in ST mating contexts [contrary to the author’s prediction]. If the hypothesis is #4, then women might elevate the importance of status in a potential mate. The key point is that the author’s predictions about sex differences in ST mating contexts are problematic and not well grounded theoretically. They really need to be clarified for coherent predictions to be made.

It should be noted that, empirically, it is the case that (in delimited cultures thus far studied), women do elevate the importance they attach to physical attractiveness in ST mating contexts (see Buss & Schmitt, 1993 for the first documentation of this finding).

Third, the use of 'social status' as the sole operationalization is problematic. Although some others have used this, the 'status' variable is highly problematic, in part because it is used in different senses in different cultures, and is inherently a vague construct unless is defined explicitly. Apropos this point, Buss (1989) found universal or near universal sex differences in the importance attached to 'good financial prospects,' but not universal sex differences in the importance attached to 'status' [although most cultures did show a sex difference in the predicted direction]. Because, as the author's state, 'financial resources' is much closer to the theoretical variable of interest than 'status,' I recommend that the authors use 'good financial resources' as the key operationalization of that variable rather than status. That would be a much cleaner test of the sex differences prediction.

Fourth, although the budget allocation method has some advantages over rating scales, it also has disadvantages. By being a forced choice procedure, it preclude discovering many patterns of results that might be present (e.g., if people valued two qualities highly, and placed no value on the other qualities). That is the reason that Buss (1989) used two different methods – a rating method AND a ranking method, which is structurally nearly identical to the 'budget allocation' method in that it forces participants to make tradeoffs. Both (budget allocation and ranking) are forced choice methods and have the same strengths and limitations.

Fifth, given the modest scope of the proposed study, I see no reason why the authors could not include a measure of age preference in a mate, or age differences preferred between self and mate. This would increase the value of the study in at least two ways. First, it would provide a critical test of a key hypothesis about sex differences in mate preferences – with men predicted to prefer younger partners than do women. Second, given the LT v. ST design, it would allow a test of the prediction that men shift their age preferences to be somewhat younger in LT mating contexts (i.e., women of higher reproductive value or future reproductive potential, perhaps late teens) and slightly higher in ST mating contexts (i.e., closer to peak fertility, which would be mid-twenties). So far, this latter hypothesis has not been sufficiently tested to determine whether it has any empirical support. This would add greatly to the novelty of the current proposed research.

These are the main suggestions. However, I'll add a minor one. It is my understanding that one of the authors has criticized mate preference research as being too heavily focused on heterosexual populations. Thus, I was surprised that the current study did not include a sample of non-heterosexual participants. Perhaps the proposed N in the current studies is too modest to include large enough samples of the latter, but given the relatively brief procedures proposed, I don't see any reason why the sample size could not be expanded.

In short, I think the study design and theoretical rationale really need to be improved in order for the study to make a novel and high quality contribution to the large extant scientific literature on human mate preferences.

Author's Response to Decision Letter for (RSOS-181243.R0)

See Appendix B.

RSOS-181243.R1 (Revision)

Review form: Reviewer 1

Is the language acceptable?

Yes

Do you have any ethical concerns with this paper?

No

Have you any concerns about statistical analyses in this paper?

No

Recommendation?

Accept with minor revision

Comments to the Author(s)

The authors have briefly but sufficiently addressed the main concerns raised in the original review. The one thing I would suggest noting in the paper (mentioned in the first review) is that with the budget allocation method, people's allocations represent not only their trait preferences but their trait priorities (given the budget constraint and that they are forced to choose among relatively low overall trait levels). Otherwise, this is acceptable.

Review form: Reviewer 2

Is the language acceptable?

Yes

Do you have any ethical concerns with this paper?

No

Have you any concerns about statistical analyses in this paper?

No

Recommendation?

Accept with minor revision

Comments to the Author(s)

The authors successfully addressed most of my concerns. I still, however, have a few comments.

My concern about the potential differences in the interpretation of the traits between the two groups was only partially solved. I like the manipulation check added at the end of the survey. However, I reiterate my suggestion to add step 4 of the translation process for the English version too (and not only for the Chinese version). Thus, at the end, two groups of native speakers of EACH language (N=2*10) will be asked to explain the different traits and to give synonyms for each one. Then, the answers for the two groups would be compared to make sure that the two groups are rating the same concepts.

I find it surprising that the authors prefer to build their predictions from the results of one study rather than from theory, but I can understand it in the context of a replication study. However, more clarity about this choice is needed. For example, I would make it clear when prediction 1b is introduced: “this sex difference will be significantly more pronounced when choosing for potential short-term partners than long-term partners (Prediction 1b, WHICH IS IN CONTRADICTION WITH THEORY, BUT CONCORDANT WITH LI ET AL’S RESULTS).”

Similarly, I would point out this discrepancy at the end of page 3: “CONTRARY TO THE THEORY, this sex difference in preference for physical attractiveness was particularly pronounced when participants were choosing for potential short-term partners.”

I think these two additions are crucial to understand the logic of the study. Otherwise it seems that the authors are making the wrong predictions, or that they made a mistake when mentioning the theory.

The following previous comment was not addressed by the authors:

I do not understand the theory behind prediction 3: “Chinese women will allocate significantly more mate dollars to social status than UK women will.” To support this prediction, the authors state that “Li et al. [5] suggested this latter result was consistent with social status being more important for social interactions generally in Eastern than Western cultures [6]”.

This does not explain why this is only found for women. Moreover, it would be nice to give more theoretical background on why social status is more important in Eastern than Western cultures. If the authors do not have any theoretical reason to backup this prediction and only want to replicate Li et al’s results, this is fine, but it should be clearly stated.

Following one of the reviewers’ comment, the authors replaced the item “social status” with “social status (i.e., good financial resources)”. I am wondering if this could be an issue, as it is not a perfect replication of Li et al.’s anymore (consequently, it could be argued that the different results could be explained by this different item). If this is not an exact replication anymore, maybe the authors should also add some clarification comments for the other items (maybe the synonyms found during the “external readings” step). This may help with the translation issue, as each item would be more extensively described. I suppose the authors have to choose between an exact replication and a different (but possibly improved) replication.

Finally, the authors improved the introduction, but one sentence is still unclear (page 2, lines 30 to 35). I would write: “because fertility declines faster with age AND REQUIRES A LARGER PHYSIOLOGICAL COST FOR women than men, men are hypothesized to show stronger preferences for PHYSICAL cues of reproductive capacity (e.g., youth, health, and good nutritional status) in women than women do when assessing the attractiveness of potential mates”.

Review form: Reviewer 4

Is the language acceptable?

Yes

Do you have any ethical concerns with this paper?

No

Have you any concerns about statistical analyses in this paper?

No

Recommendation?

Accept in principle

Comments to the Author(s)

The authors have been responsive to suggestions, and the current research proposal is acceptable.

Decision letter (RSOS-181243.R1)

09-Oct-2018

Dear Dr Jones,

The Subject Editor assigned to your paper ("Chinese and UK participants' preferences for physical attractiveness and social status in potential mates (Stage 1 Registered Report)") has now received comments from reviewers. We would like you to revise your paper in accordance with the referee and Associate Editor suggestions which can be found below (not including confidential reports to the Editor).

Please submit a copy of your revised paper before 17-Oct-2018.

- Ethics statement

- Data accessibility

It is a condition of publication that all supporting data are made available either as supplementary information or preferably in a suitable permanent repository. The data accessibility section should state where the article's supporting data can be accessed. This section should also include details, where possible of where to access other relevant research materials such as statistical tools, protocols, software etc can be accessed. If the data have been deposited in an external repository, this section should list the database, accession number and link to the DOI for all data from the article that have been made publicly available. Data sets that have been

deposited in an external repository and have a DOI should also be appropriately cited in the manuscript and included in the reference list.

- **Competing interests**

- **Authors' contributions**

- **Acknowledgements**

- **Funding statement**

Please note that Royal Society Open Science charge article processing charges for all new submissions that are accepted for publication. Charges will also apply to papers transferred to Royal Society Open Science from other Royal Society Publishing journals, as well as papers submitted as part of our collaboration with the Royal Society of Chemistry (<http://rsos.royalsocietypublishing.org/chemistry>). If your manuscript is newly submitted and subsequently accepted for publication, you will be asked to pay the article processing charge, unless you request a waiver and this is approved by Royal Society Publishing. You can find out more about the charges at <http://rsos.royalsocietypublishing.org/page/charges>. Should you have any queries, please contact openscience@royalsociety.org.

Kind regards,

Royal Society Open Science Editorial Office
Royal Society Open Science
openscience@royalsociety.org

on behalf of Chris Chambers (Subject Editor)
openscience@royalsociety.org

Associate Editor Comments to Author (Professor Chris Chambers):

The manuscript was returned to three of the four original reviewers. All are positive overall, but Reviewer 1 and 2 recommend some important remaining amendments to the framing of the hypotheses, procedures, and clarity in key places. Provided the revision addresses these points comprehensively, IPA should be forthcoming without requiring further in-depth review.

Reviewer comments to Author:

Reviewer: 1

Comments to the Author(s)

The authors have briefly but sufficiently addressed the main concerns raised in the original review. The one thing I would suggest noting in the paper (mentioned in the first review) is that with the budget allocation method, people's allocations represent not only their trait preferences but their trait priorities (given the budget constraint and that they are forced to choose among relatively low overall trait levels). Otherwise, this is acceptable.

Reviewer: 2

Comments to the Author(s)

The authors successfully addressed most of my concerns. I still, however, have a few comments.

My concern about the potential differences in the interpretation of the traits between the two groups was only partially solved. I like the manipulation check added at the end of the survey. However, I reiterate my suggestion to add step 4 of the translation process for the English version too (and not only for the Chinese version). Thus, at the end, two groups of native speakers of EACH language (N=2*10) will be asked to explain the different traits and to give synonyms for each one. Then, the answers for the two groups would be compared to make sure that the two groups are rating the same concepts.

I find it surprising that the authors prefer to build their predictions from the results of one study rather than from theory, but I can understand it in the context of a replication study. However, more clarity about this choice is needed. For example, I would make it clear when prediction 1b is introduced: "this sex difference will be significantly more pronounced when choosing for potential short-term partners than long-term partners (Prediction 1b, WHICH IS IN CONTRADICTION WITH THEORY, BUT CONCORDANT WITH LI ET AL'S RESULTS)."

Similarly, I would point out this discrepancy at the end of page 3: "CONTRARY TO THE THEORY, this sex difference in preference for physical attractiveness was particularly pronounced when participants were choosing for potential short-term partners."

I think these two additions are crucial to understand the logic of the study. Otherwise it seems that the authors are making the wrong predictions, or that they made a mistake when mentioning the theory.

The following previous comment was not addressed by the authors:

I do not understand the theory behind prediction 3: "Chinese women will allocate significantly more mate dollars to social status than UK women will." To support this prediction, the authors state that "Li et al. [5] suggested this latter result was consistent with social status being more important for social interactions generally in Eastern than Western cultures [6]".

This does not explain why this is only found for women. Moreover, it would be nice to give more theoretical background on why social status is more important in Eastern than Western cultures. If the authors do not have any theoretical reason to backup this prediction and only want to replicate Li et al's results, this is fine, but it should be clearly stated.

Following one of the reviewers' comment, the authors replaced the item "social status" with "social status (i.e., good financial resources)". I am wondering if this could be an issue, as it is not a perfect replication of Li et al.'s anymore (consequently, it could be argued that the different results could be explained by this different item). If this is not an exact replication anymore, maybe the authors should also add some clarification comments for the other items (maybe the synonyms found during the "external readings" step). This may help with the translation issue, as each item would be more extensively described. I suppose the authors have to choose between an exact replication and a different (but possibly improved) replication.

Finally, the authors improved the introduction, but one sentence is still unclear (page 2, lines 30 to 35). I would write: "because fertility declines faster with age AND REQUIRES A LARGER PHYSIOLOGICAL COST FOR women than men, men are hypothesized to show stronger preferences for PHYSICAL cues of reproductive capacity (e.g., youth, health, and good nutritional status) in women than women do when assessing the attractiveness of potential mates".

Reviewer: 4

Comments to the Author(s)

The authors have been responsive to suggestions, and the current research proposal is acceptable.

Author's Response to Decision Letter for (RSOS-181243.R1)

See Appendix C.

Decision letter (RSOS-181243.R2)

18-Oct-2018

Dear Dr Jones

On behalf of the Editor, I am pleased to inform you that your Manuscript RSOS-181243.R2 entitled "Chinese and UK participants' preferences for physical attractiveness and social status in potential mates (Stage 1 Registered Report)" has been accepted in principle for publication in Royal Society Open Science. The reviewers' and editors' comments are included at the end of this email.

You may now progress to Stage 2 and complete the study as approved. Before commencing data collection we ask that you:

- 1) Update the journal office as to the anticipated completion date of your study.

2) Register your approved protocol on the Open Science Framework (<https://osf.io/>) or other recognised repository, either publicly or privately under embargo until submission of the Stage 2 manuscript. Please note that a time-stamped, independent registration of the protocol is mandatory under journal policy, and manuscripts that do not conform to this requirement cannot be considered at Stage 2. The protocol should be registered unchanged from its current approved state, with the time-stamp preceding implementation of the approved study design.

Following completion of your study, we invite you to resubmit your paper for peer review as a Stage 2 Registered Report. Please note that your manuscript can still be rejected for publication at Stage 2 if the Editors consider any of the following conditions to be met:

- The results were unable to test the authors' proposed hypotheses by failing to meet the approved outcome-neutral criteria.
- The authors altered the Introduction, rationale, or hypotheses, as approved in the Stage 1 submission.
- The authors failed to adhere closely to the registered experimental procedures. Please note that any deviations from the approved experimental procedures must be communicated to the editor immediately for approval, and prior to the completion of data collection. Failure to do so can result in revocation of in-principle acceptance and rejection at Stage 2 (see complete guidelines for further information).
- Any post-hoc (unregistered) analyses were either unjustified, insufficiently caveated, or overly dominant in shaping the authors' conclusions.
- The authors' conclusions were not justified given the data obtained.

We encourage you to read the complete guidelines for authors concerning Stage 2 submissions at <http://rsos.royalsocietypublishing.org/content/registered-reports>. Please especially note the requirements for data sharing, reporting the URL of the independently registered protocol, and that withdrawing your manuscript will result in publication of a Withdrawn Registration.

Please note that Royal Society Open Science will introduce article processing charges for all new submissions received from 1 January 2018. Registered Reports submitted and accepted after this date will ONLY be subject to a charge if they subsequently progress to and are accepted as Stage 2 Registered Reports. If your manuscript is submitted and accepted for publication after 1 January 2018 (i.e. as a full Stage 2 Registered Report), you will be asked to pay the article processing charge, unless you request a waiver and this is approved by Royal Society Publishing. You can find out more about the charges at <http://rsos.royalsocietypublishing.org/page/charges>. Should you have any queries, please contact openscience@royalsociety.org.

Once again, thank you for submitting your manuscript to Royal Society Open Science and we look forward to receiving your Stage 2 submission. If you have any questions at all, please do not hesitate to get in touch. We look forward to hearing from you shortly with the anticipated submission date for your stage two manuscript.

Kind regards,

Royal Society Open Science Editorial Office
Royal Society Open Science
openscience@royalsociety.org

on behalf of Professor Chris Chambers (Registered Reports Editor, Royal Society Open Science)
openscience@royalsociety.org

Author's Response to Decision Letter for (RSOS-181243.R2)

See Appendix D.

RSOS-181243.R3 (Revision)

Review form: Reviewer 2

Is the language acceptable?

Yes

Do you have any ethical concerns with this paper?

No

Have you any concerns about statistical analyses in this paper?

Yes

Recommendation?

Accept with minor revision

Comments to the Author(s)

- The data is able to test the authors' proposed hypotheses.
- The introduction, rationale and stated hypotheses are the same as the approved Stage 1 submission.
- The authors adhered to the registered experimental procedures, except for the number or participants recruited (higher than planned). The authors are honest and open about this mistake, and I do not think this is a major issue for the paper. However, I think it would be a nice addition for the paper if the authors could show that the results are the same when using the pre-registered number of participants (by randomly removing some participants in each group to match the registered experimental procedure).
- One missing element is any attempt to show that the two groups being compared (UK and Chinese) are actually comparable. I know that the authors did not collect any demographic information apart from age (which is an issue when comparing cross-cultural groups), but the authors could at least make sure that there is no significant age difference between the groups being compared (as age can change mating preferences).
- I have one concern about the statistical analyses of the explanatory analysis. The authors are not controlling for multi-testing as they should, and some of their results would not be significant after a simple Bonferroni correction. For example (but not limited to): effect of participant sex in the Chinese sample for creativity and the interaction sex/relationship context for kindness in the UK sample.

- There is one result that is in contradiction with the previous literature (and with the theory), that the authors are not discussing at all: it seems that male participants prefer an older partner than themselves for a long-term relationship (unless I misunderstood the methods). Given that this result is surprising, I think that one sentence explaining or at least acknowledging it would be useful.
- The authors could add the number of participants (after exclusion) in each cell of the 6 tables presenting the results to increase clarity.
- I really like the plots showing the distribution of scores. I was wondering if it would be possible to find a way to add the different means on the plots without diminishing their readability. I would also like to see the same graphs for the other variables (creativity, kindness, liveliness and maybe age) in the supplementary.

Review form: Reviewer 3

Is the language acceptable?

Yes

Do you have any ethical concerns with this paper?

No

Have you any concerns about statistical analyses in this paper?

No

Recommendation?

Accept with minor revision

Comments to the Author(s)

This registered report (stage 2) deals with the question whether cross cultural (sex) differences in mate preferences are robust. The planned study is a replication of Li and colleagues (2011; PAID). The authors followed their registered procedure and transparently state all deviations (e.g. recruiting more participants than registered). They also report very interesting exploratory results. I really appreciate their open data and code and I especially liked the described translation process. I think this study will make a good contribution to the literature. Below I outline rather minor points that might help to improve the manuscript.

1. I got a bit confused by some of the used sub-headlines. More precisely, the authors use "Testing Prediction 1" on page 8 and page 11. The same is true for testing the other predictions. The authors might want to change one of the sub-headlines.
2. The exclusion rate seems rather high (especially for testing prediction 1). Can the authors give more information on how many participants got excluded for which reasons (e.g. how many failed the manipulation check, how many were outliers etc.)?
3. Regarding the exclusion rate again, the authors could think about computing robustness checks including all participants to see if there are any differences in results.
4. Please report effect sizes and/ or confidence intervals for all effects.
5. Regarding the exploratory results section, can you add a sentence (to all results reported here) about the direction of the effect, as you did in the main analyses?
6. Do I understand it correctly, that men preferred slightly older women for long-term relationships (Table 6)? If yes, this is surprising. The authors may want to mention/ discuss this finding.

Decision letter (RSOS-181243.R3)

22-Aug-2019

Dear Dr Jones:

On behalf of the Editor, I am pleased to inform you that your Stage 2 Registered Report RSOS-181243.R3 entitled "Chinese and UK participants' preferences for physical attractiveness and social status in potential mates (Stage 2 Registered Report)" has been deemed suitable for publication in Royal Society Open Science subject to minor revision in accordance with the referee suggestions. Please find the referees' comments at the end of this email.

The reviewers and Subject Editor have recommended publication, but also suggest some minor revisions to your manuscript. Therefore, I invite you to respond to the comments and revise your manuscript.

Please also ensure that all the below editorial sections are included where appropriate -- if any section is not applicable to your manuscript, please can we ask you to nevertheless include the heading, but explicitly state that the heading is inapplicable. An example of these sections is attached with this email.

- Ethics statement

- Data accessibility

[http://datadryad.org/submit?journalID=RSOS&manu=\(Document not available\)](http://datadryad.org/submit?journalID=RSOS&manu=(Document not available))

- Competing interests

- Authors' contributions

- Acknowledgements

- Funding statement

Because the schedule for publication is very tight, it is a condition of publication that you submit the revised version of your manuscript within 7 days (i.e. by the 30-Aug-2019). If you do not think you will be able to meet this date please let me know immediately.

Supplementary files will be published alongside the paper on the journal website and posted on the online figshare repository (<https://figshare.com>). The heading and legend provided for each

supplementary file during the submission process will be used to create the figshare page, so please ensure these are accurate and informative so that your files can be found in searches. Files on figshare will be made available approximately one week before the accompanying article so that the supplementary material can be attributed a unique DOI.

Please note that Royal Society Open Science will introduce article processing charges for all new submissions received from 1 January 2018. Registered Reports submitted and accepted after this date will ONLY be subject to a charge if they subsequently progress to and are accepted as Stage 2 Registered Reports. If your manuscript is submitted and accepted for publication after 1 January 2018 (i.e. as a full Stage 2 Registered Report), you will be asked to pay the article processing charge, unless you request a waiver and this is approved by Royal Society Publishing. You can find out more about the charges at <http://rsos.royalsocietypublishing.org/page/charges>. Should you have any queries, please contact openscience@royalsociety.org.

on behalf of Professor Chris Chambers
(Registered Reports Editor, Royal Society Open Science)
openscience@royalsociety.org

Associate Editor Comments to Author (Professor Chris Chambers):

The Stage 2 manuscript was returned to two of the four reviewers who assessed the protocol at Stage 1. Both are positive about the submission and offer some suggestions for minor revisions, including clarification of the exploratory analyses (and consideration of multiple comparisons), consideration of robustness checks, and reporting of descriptive statistics. Provided the authors are able to respond comprehensively to these points (via revision or rebuttal), final Stage 2 acceptance should be forthcoming without requiring further in-depth review.

Comments to Author:

Reviewer: 2

Comments to the Author(s)

- The data is able to test the authors' proposed hypotheses.
- The introduction, rationale and stated hypotheses are the same as the approved Stage 1 submission.
- The authors adhered to the registered experimental procedures, except for the number or participants recruited (higher than planned). The authors are honest and open about this mistake, and I do not think this is a major issue for the paper. However, I think it would be a nice addition

for the paper if the authors could show that the results are the same when using the pre-registered number of participants (by randomly removing some participants in each group to match the registered experimental procedure).

- One missing element is any attempt to show that the two groups being compared (UK and Chinese) are actually comparable. I know that the authors did not collect any demographic information apart from age (which is an issue when comparing cross-cultural groups), but the authors could at least make sure that there is no significant age difference between the groups being compared (as age can change mating preferences).
- I have one concern about the statistical analyses of the explanatory analysis. The authors are not controlling for multi-testing as they should, and some of their results would not be significant after a simple Bonferroni correction. For example (but not limited to): effect of participant sex in the Chinese sample for creativity and the interaction sex/relationship context for kindness in the UK sample.
- There is one result that is in contradiction with the previous literature (and with the theory), that the authors are not discussing at all: it seems that male participants prefer an older partner than themselves for a long-term relationship (unless I misunderstood the methods). Given that this result is surprising, I think that one sentence explaining or at least acknowledging it would be useful.
- The authors could add the number of participants (after exclusion) in each cell of the 6 tables presenting the results to increase clarity.
- I really like the plots showing the distribution of scores. I was wondering if it would be possible to find a way to add the different means on the plots without diminishing their readability. I would also like to see the same graphs for the other variables (creativity, kindness, liveliness and maybe age) in the supplementary.

Reviewer: 3

Comments to the Author(s)

This registered report (stage 2) deals with the question whether cross cultural (sex) differences in mate preferences are robust. The planned study is a replication of Li and colleagues (2011; PAID). The authors followed their registered procedure and transparently state all deviations (e.g. recruiting more participants than registered). They also report very interesting exploratory results. I really appreciate their open data and code and I especially liked the described translation process. I think this study will make a good contribution to the literature. Below I outline rather minor points that might help to improve the manuscript.

1. I got a bit confused by some of the used sub-headlines. More precisely, the authors use "Testing Prediction 1" on page 8 and page 11. The same is true for testing the other predictions. The authors might want to change one of the sub-headlines.
2. The exclusion rate seems rather high (especially for testing prediction 1). Can the authors give more information on how many participants got excluded for which reasons (e.g. how many failed the manipulation check, how many were outliers etc.)?
3. Regarding the exclusion rate again, the authors could think about computing robustness checks including all participants to see if there are any differences in results.
4. Please report effect sizes and/ or confidence intervals for all effects.
5. Regarding the exploratory results section, can you add a sentence (to all results reported here) about the direction of the effect, as you did in the main analyses?

6. Do I understand it correctly, that men preferred slightly older women for long-term relationships (Table 6)? If yes, this is surprising. The authors may want to mention/ discuss this finding.

Author's Response to Decision Letter for (RSOS-181243.R3)

See Appendix E.

Decision letter (RSOS-181243.R4)

23-Oct-2019

Dear Dr Jones:

It is a pleasure to accept your manuscript entitled "Chinese and UK participants' preferences for physical attractiveness and social status in potential mates (Stage 2 Registered Report)" in its current form for publication in Royal Society Open Science.

Kind regards,

on behalf of Professor Chris Chambers (Subject Editor)
openscience@royalsociety.org

Dear Editors,

Here we submit a manuscript to be considered as a Stage 1 Registered Report. This is a revised version of RSOS-181000.

Are there robust sex differences in human mate preferences? Men are hypothesized to show stronger preferences for physical attractiveness in potential mates than women are, particularly when assessing the attractiveness of potential mates for short-term relationships. By contrast, women are thought to show stronger preferences for social status in potential mates than men are, particularly when assessing the attractiveness of potential mates for long-term relationships. Although these mate-preference sex differences are often claimed to be ‘universal’ (i.e., stable across cultures), evidence for cross-cultural similarities in these mate-preference sex differences is mixed and somewhat controversial. Consequently, we will use an established “budget allocation” task to investigate Chinese and UK participants’ preferences for physical attractiveness and social status in potential mates.

We confirm that all necessary support (e.g. funding, facilities) and approvals (e.g. ethics) are in place for the proposed research. We agree to register the approved protocol on the OSF as soon as it is approved, along with all analysis code and materials. We also agree to make all data publicly available on the OSF once it is collected. Data collection and analyses will be completed within 9 months of approval of the protocol. If we withdraw our paper after provisional acceptance, we agree to the journal publishing a short summary of the pre-registered study under a section Withdrawn Registrations.

We suggest the following researchers as particularly appropriate referees. Each has published extensively on human mate preferences.

Jeanne Bovet, Institute of Advanced Studies at Toulouse,
jeanne.bovet@iast.fr

Julia Juenger, University Goettingen, julia.juenger@psych.uni-goettingen.de

We thank Chris Chambers for his thoughtful and constructive comments on our original submission. We have addressed each of these in our manuscript and detail below the changes that we have made. We strongly believe that addressing his comments has strengthened our submission considerably.

1. Power analysis/sampling plan. Where authors propose frequentist analytic methods, all hypothesis tests should be associated with a power analysis to detect either (a) the smallest effect of theoretical interest, or (b) the lower bound estimate of the expected effect size from a representative sample of the relevant prior literature. This condition is not yet met in your proposal. As an alternative to frequentist testing, you may wish to consider alternative Bayesian inferential methods that permit optional stopping to maximise the efficiency of sampling and to also permit positive conclusions about negative

results. For further information, please see Dienes 2014
<http://journal.frontiersin.org/article/10.3389/fpsyg.2014.00781/full>

We now report power analyses describing the smallest effect size we have 90% power to detect for each critical test (see pages 7 and 8).

2. Please ensure that there is a direct correspondence between the proposed hypotheses, the power analyses (or alternative sampling plans), and the proposed statistical tests. At present this connection is not sufficiently explicit to proceed to in-depth review. The power analysis needs to map directly on to the proposed statistical tests, each of which need to correspond to one of the specified hypotheses. For maximum clarity, we recommend numbering the hypotheses at the end of the Introduction (e.g. in a list, as you have done) and then similarly listing in the analysis plan which statistical test addresses which specific hypothesis, using which dependent variables, and reporting how each test or test component achieves the designated power level (we recommend achieving a minimum of 90% but a lower level can be proposed; where including power analysis please be sure to include and justify all input parameters, such as for ANOVA the assumed correlation between repeated measures where applicable, and the method used for estimation).

We now detail our specific (i.e., numbered) predictions, the specific tests that will be used for each, and the minimum effect size each test can detect with 90% power (see pages 7 and 8).

3. Please ensure that all procedures are described in sufficient detail to enable direct replication without the reader needing to review prior work. For instance, how are the various response categories – attractiveness, creativity etc – presented to participants? Perhaps provide a visual schematic of the procedure, details of translations etc. Registered Reports must be internally reproducible the maximum reasonable extent.

We have included a screen grab of the interface we will use to collect responses (see page 5).

4. One of the key criteria that reviewers are asked to assess in Stage 1 RRs is "Whether the authors have considered sufficient outcome-neutral conditions (e.g. absence of floor or ceiling effects; positive controls; other quality checks) for ensuring that the results obtained are able to test the stated hypotheses.", and successfully passing such tests is an editorial criterion at Stage 2 following completion of the study. Your protocol does not seem to propose any such tests - therefore please consider whether such positive controls or data quality checks are necessary (they might not be for some designs), and if possible how they might be included in the design. Such tests should be orthogonal of the main hypotheses and, where they involve a hypothesis test, should be accompanied by a power analysis or alternative (e.g. Bayesian) sampling plan. We recommend outlining any such tests in a separate section.

We now detail all data exclusions. No other data quality checks will be applied (see page 8).

5. Please ensure that exclusion criteria for data within and across participants are comprehensively pre-specified as it usually not possible to adjust these for pre-registered analyses after provisional acceptance is granted. For example, you note that “Responses more than three standard deviations from the mean for the sample will be excluded from the dataset prior to analyses” (p5). What does “the sample” refer to? The total sample? Each national sample separately? For which DVs or averaged across which DVs? Within which conditions or sub-conditions?

We have clarified these points in our manuscript on page 8.

6. Please provide details of ethical approval in the main text of the manuscript.

We have added this information to our manuscript on page 4.

7. Does recruitment involve any screening for age, sexual orientation, or ethnic status within each country? Please explain in the manuscript.

We have clarified these points in our manuscript on page 4.

8. Given that all hypotheses relate to social status and physical attractiveness, what is the rationale for including creativity, kindness, and liveliness as measures? Are these for exploratory analysis only?

We have clarified these points in our manuscript on page 9.

9. In general, to what extent might cultural differences in outcomes be explained by slightly different definitions or conceptualisations of the terms between languages? Do you have confirmation, for instance, that complex concepts like “social status” are understood and conceptualised identically between countries?

We have detailed that we will use the Psychological Science Accelerator’s translation protocols to ensure nuance of terms is preserved in translated versions (see pages 5 and 6).

We look forward to hearing from you in due course.

Sincerely,

Benedict Jones & Lingshan Zhang

We thank each of the reviewers for their thoughtful and constructive comments on our submission. We have detailed the changes we have made below. We agree that addressing these comments has strengthened our manuscript considerably.

Reviewer 1's comments

1. Reviewer 1 asked that we elaborate on the methodological problems with rating and ranking paradigms that the budget-allocation paradigm addresses (and outlined arguments we should make).

We now describe the specific methodological problem that the budget-allocation paradigm addresses.

Text added to page 3: These paradigms can be problematic because trait ratings do not require participants to trade off traits against each other and because trait rankings do not contain information about the relative strength of preference for traits [6,8].

2. Reviewer 1 asked that we clarify that the budget allocation paradigm allows participants to purchase percentile points on each dimension, with each dollar corresponding to a percentile point.

We have clarified this point in our Methods.

Text added to page 5: On-screen instructions will inform participants that each dollar corresponds to a percentile point on that trait.

3. Reviewer 1 notes that we could also use these data to test whether men value attractiveness in both short and long term contexts, while women value attractiveness more than other traits in short term, but not long term, contexts.

We now note that we will report such tests in our Exploratory Analyses section.

Text added to page 10: Exploratory analyses testing whether women value physical attractiveness more than other traits for short-term, but not long-term, relationships, while men value physical attractiveness more than other traits for both short- and long-term relationships will also be reported in this section, along with exploratory analyses testing for cultural and sex differences in age preferences [5].

Reviewer 2's comments

1. Reviewer 2 asked that we clarify that predictions 1b and 3 may not necessarily be straightforward predictions from theory, but are Li et al's results.

We have clarified that all of our predictions are based on Lie et al's results.

Text added to page 4: In the current study, we will attempt to replicate three key results from Li et al [6]. By contrast with Li et al [6], who reported these results for US and Singaporean participants, we will attempt to replicate their key results in UK and Chinese participants.

2. Reviewer 2 suggested an addition to the PSA translation procedure and also noted how this could be used to generate material for a manipulation check.

We have made this change.

Text added to page 6: After completing the budget-allocation task, participants will be asked to complete a manipulation-check task to ensure they understood what each trait represented (see Data exclusions section, below) and to report the age of their ideal long-term and short-term partner. These age-preference data will be used in exploratory analyses testing for cultural and sex differences in age preferences.

Text added to page 7: External Readers: Will read materials for final clarity check (N=10, all non-academics).

Text added to page 7: Step 4 (External readings). Version C is tested on ten non-academics fluent in the target language. Members of the fluent group are asked how they perceive and understand the translation. Possible misunderstandings are noted and again discussed as in Step 3. Note that the Psychological Science Accelerator's procedures for translation use two, rather than ten, bilingual speakers in Step 4.

Text added to page 9: At Step 4 of the translation process, the external speakers will be asked to agree on three synonyms for each of the to be traits tested. Participants will be asked to match these synonyms to the traits at the end of the study. Participants who fail this manipulation-check task for any traits will be excluded from analyses.

3. Reviewer 2 asked that we clarify the recruitment criteria for Chinese and UK participants.

We have clarified this in our Methods.

Text added to page 5: Only participants between the ages of 16 and 30 years of age born in either China (Chinese participants) or the UK (UK participants) will be recruited.

4. Reviewer 2 asked that we note health and nutritional status as additional factors that influence reproductive capacity and clarify that pregnant and lactating women have both greater need for resources and reduced ability to obtain resources.

We have made these changes.

Text added to page 2: First, because fertility declines faster with age in women than men, men are hypothesized to show stronger preferences for cues of reproductive capacity (e.g., youth, health, and good nutritional status) in women than women do when assessing the attractiveness of potential mates [1]. Second, women bear greater costs of obligatory parental investment (i.e., pregnancy and lactation) than men do, meaning they have both a greater need for resources and reduced ability to obtain resources [2].

5. Reviewer 2 suggested that we mention that we will only use the other traits for exploratory analyses earlier in the manuscript (e.g., in the methods).

We have made this change.

Text added to page 5: Data for traits other than attractiveness and social status will be reported in an exploratory analyses section. These traits are only included in the study because they were included in Li et al. [6].

Reviewer 3's comments

1. Reviewer 3 notes that our samples were not identical to those in Li et al.

We have emphasized this point in our Introduction.

Text added to page 4: In the current study, we will attempt to replicate three key results from Li et al [6]. By contrast with Li et al [6], who reported these results for US and Singaporean participants, we will attempt to replicate their key results in UK and Chinese participants.

2. Reviewer 3 suggested also citing Li et al's earlier work using the budget allocation paradigm.

We have added this citation.

Text added to page 3: Li et al. [8] developed the budget-allocation task to address the methodological limitations of trait-rating and -ranking paradigms.

3. Reviewer 3 suggested we include more discussion of the limitations of alternative methods.

We have clarified the major problem addressed by the budget-allocation task.

Text added to page 3: These paradigms can be problematic because trait ratings do not require participants to trade off traits against each other and because trait rankings do not contain information about the relative strength of preference for traits [6,8].

4. Reviewer 3 suggested there was no need to restrict UK participants to white participants and that we should clarify our recruitment criteria (that is, how we define Chinese and UK participants).

We agree and have clarified that Chinese participants will be defined as being born in China and UK participants defined as being born in the UK.

Text added to page 5: Only participants between the ages of 16 and 30 years of age born in either China (Chinese participants) or the UK (UK participants) will be recruited.

5. Reviewer 3 caught that prediction 3 should be for long-term relationships only.

We are grateful to Reviewer 3 for catching this error on our part and we have clarified this point throughout. We have also amended our analysis code accordingly.

Text added to page 4: When choosing for potential long-term partners, Chinese women will allocate significantly more mate dollars to social status than UK women will.

6. Reviewer 3 suggested increasing our sample size to 125 per group.

We have made this change.

7. Reviewer 3 asked what other demographic variables we will collect.

We have clarified that, other than age, will we collect no further demographic information from participants.

Text added to page 5: Other than age, will we collect no further demographic information from participants.

8. Reviewer 3 asked how the effect size in our power analyses matches against those in the literature.

We have sufficient power to detect small effect sizes in our confirmatory analyses. As Reviewer 4 notes, these effects are typically large.

Reviewer 4's comments

1. Reviewer 4 noted that there is good evidence for robust sex differences in long-term preferences.

We have clarified this point in our Introduction.

Text added to page 3: Indeed, there is good evidence that these mate-preference sex differences do occur in diverse cultures, at least when people express preferences for long-term relationships, such as marriage (e.g., [7]). While evidence for cross-cultural similarity in mate-preference sex differences for long-term relationships is well established, fewer studies have investigated mate-preference sex differences for short-term relationships.

2. Reviewer 4 disagrees with some of the rationale for the predictions and noted that Buss and Schmidt 1993 discuss alternative models for sex differences in mate preferences.

We have clarified that alternative perspectives exist in our Introduction.

Text added to page 2: Consistent with these hypotheses (see [5] for additional theoretical perspectives), studies have reported that women place greater emphasis on social status (i.e., resources) and men place greater emphasis on physical attractiveness when assessing potential long-term partners, while both men and women place great emphasis on physical attractiveness when assessing potential short-term partners [reviewed in 6].

3. Reviewer 4 suggested replacing social status with good financial resources.

We have added a parenthetical statement making this point.

Text added to page 5: In this task, participants are instructed to distribute a total budget of 100 mate dollars across each of the following traits to choose a hypothetical partner; physical attractiveness, social status (i.e., good financial resources), creativity, kindness, and liveliness.

4. Reviewer 4 argued that the ranking method used in previous Buss studies has the same limitations as the budget allocation method.

We respectfully disagree with this point and note that Reviewer 1 highlighted a methodological weakness of the ranking method that the budget-allocation method addresses.

Text added to page 3: Li et al. [8] developed the budget-allocation task to address the methodological limitations of trait-rating and -ranking paradigms.

5. Reviewer 4 suggested adding an ideal age preference test and suggested his would greatly increase the novelty of the study.

We now note that we will collect this data (at the end of the study – i.e., after the main test) and report analyses of these data in our exploratory analyses section.

Text added to page 6: After completing the budget-allocation task, participants will be asked to complete a manipulation-check task to ensure they understood what each trait represented (see Data exclusions section, below) and to report the age of their ideal long-term and short-term partner. These age-preference data will be used in exploratory analyses testing for cultural and sex differences in age preferences.

Text added to page 10: Exploratory analyses testing whether women value physical attractiveness more than other traits for short-term, but not long-term, relationships, while men value physical attractiveness more than other traits

for both short- and long-term relationships will also be reported in this section, along with exploratory analyses testing for cultural and sex differences in age preferences [5].

6. Reviewer 4 noted that one of the authors has criticized the focus on heterosexual mate preferences in the evolutionary psychology literature and suggested queer mate preferences could be assessed easily in the current study.

As far as we can tell, the only public comment on this issue that any of the authors have made were tweets by Ben Jones suggesting that practices like the emphasis on heterosexual preferences in the mate preference literature indicated that, although conservatives may be underrepresented in psychology departments, non-traditional perspectives and experiences were still underrepresented in psychological research and theories. We disagree it would be straightforward to address this issue in the current study. We would argue instead that this point actually supports the claim that Western values have a strong influence on how people think about psychological science. It is only in the last 20 years that laws prohibiting sex with own-sex individuals were dropped in China and China still has no laws prohibiting discrimination against gay people. Under those circumstances, we disagree that it is straightforward to conduct a study of non-heterosexual mate preferences in China. Our lab is currently working on a large-scale study of individual differences in gay men's and women's mate preferences in a Western sample and are encouraged by Reviewer 4's interest in the issue.

We thank the reviewers for their constructive comments on our submission. We have detailed the changes we have made below.

Reviewer 1's comments

The authors have briefly but sufficiently addressed the main concerns raised in the original review. The one thing I would suggest noting in the paper (mentioned in the first review) is that with the budget allocation method, people's allocations represent not only their trait preferences but their trait priorities (given the budget constraint and that they are forced to choose among relatively low overall trait levels). Otherwise, this is acceptable.

We have added this point to our description of the budget allocation method on page three: "Li et al. [8] developed the budget-allocation task to address the methodological limitations of trait-rating and -ranking paradigms. In the budget-allocation task, participants allocate a sum from a maximum total budget of 100 mate dollars to each of the following traits in a potential partner; physical attractiveness, social status, creativity, kindness, and liveliness. Each participant performs this task twice; once when choosing for a long-term (marriage) partner and once when choosing for a short-term (casual sex) partner. Importantly, the budget-allocation task directly addresses the limitations of the trait-rating and trait-ranking tasks described above. Note that allocations represent participants' trait priorities, as well as their trait preferences."

Reviewer 2's comments

My concern about the potential differences in the interpretation of the traits between the two groups was only partially solved. I like the manipulation check added at the end of the survey. However, I reiterate my suggestion to add step 4 of the translation process for the English version too (and not only for the Chinese version). Thus, at the end, two groups of native speakers of EACH language ($N=2*10$) will be asked to explain the different traits and to give synonyms for each one. Then, the answers for the two groups would be compared to make sure that the two groups are rating the same concepts.

We have added this additional step (page 7): "Step 4 (External readings). Version C is tested on ten non-academics fluent in the target language. Members of the fluent group are asked how they perceive and understand the translation and to agree on three synonyms for each trait to be tested. Possible misunderstandings are noted and again discussed as in Step 3. A group of ten native English speakers will also be asked to agree on three synonyms for each trait to be tested. Note that the Psychological Science Accelerator's procedures for translation use two, rather than ten, bilingual speakers in Step 4."

I find it surprising that the authors prefer to build their predictions from the results of one study rather than from theory, but I can understand it in the context of a replication study. However, more clarity about this choice is needed. For example, I would make it clear when prediction 1b is introduced: "this sex difference will be significantly more pronounced when choosing for

potential short-term partners than long-term partners (Prediction 1b, WHICH IS IN CONTRADICTION WITH THEORY, BUT CONCORDANT WITH LI ET AL'S RESULTS)."

We have clarified this point in our Intro (page 4): "**Prediction 1.** Men will allocate significantly more mate dollars to physical attractiveness than women in both the UK and Chinese samples (Prediction 1a) and this sex difference will be significantly more pronounced when choosing for potential short-term partners than long-term partners (Prediction 1b). Note that, although Prediction 1b is what was reported in Li et al. [6], it is arguably inconsistent with theory [3,5]."

Similarly, I would point out this discrepancy at the end of page 3: "CONTRARY TO THE THEORY, this sex difference in preference for physical attractiveness was particularly pronounced when participants were choosing for potential short-term partners."

We have clarified this point in our Intro (pages 3 and 4): "To test for the hypothesized cross-cultural similarities in mate-preference sex differences, Li et al. [6] administered their budget-allocation task to US and Singaporean participants. Men allocated significantly more mate dollars to physical attractiveness than women did in both the US and Singaporean samples. Contrary to theory [3,5], this sex difference in preference for physical attractiveness was particularly pronounced when participants were choosing for potential short-term partners."

I think these two additions are crucial to understand the logic of the study. Otherwise it seems that the authors are making the wrong predictions, or that they made a mistake when mentioning the theory.

The following previous comment was not addressed by the authors: I do not understand the theory behind prediction 3: "Chinese women will allocate significantly more mate dollars to social status than UK women will." To support this prediction, the authors state that "Li et al. [5] suggested this latter result was consistent with social status being more important for social interactions generally in Eastern than Western cultures [6]". This does not explain why this is only found for women. Moreover, it would be nice to give more theoretical background on why social status is more important in Eastern than Western cultures. If the authors do not have any theoretical reason to backup this prediction and only want to replicate Li et al's results, this is fine, but it should be clearly stated.

We have added this point on page 4: "This sex difference in preference for social status was particularly pronounced when participants were choosing for potential long-term partners. Intriguingly, when choosing for potential long-term partners, Singaporean women allocated significantly more mate dollars to social status than US women did. Li et al. [6] suggested this latter result was consistent with social status being more important for social interactions generally in Eastern than Western cultures [9]. It is unclear, however, why this

cultural difference in preference for social status was only evident in women's preferences."

We have clarified this point further on page 5: "**Prediction 3**. When choosing for potential long-term partners, Chinese women will allocate significantly more mate dollars to social status than UK women will. Note that, although Prediction 3 is what was reported in Li et al. [6], it is unclear why this cultural difference was not also observed for men."

Following one of the reviewers' comment, the authors replaced the item "social status" with "social status (i.e., good financial resources)". I am wondering if this could be an issue, as it is not a perfect replication of Li et al.'s anymore (consequently, it could be argued that the different results could be explained by this different item). If this is not an exact replication anymore, maybe the authors should also add some clarification comments for the other items (maybe the synonyms found during the "external readings" step). This may help with the translation issue, as each item would be more extensively described. I suppose the authors have to choose between an exact replication and a different (but possibly improved) replication.

We have not made these changes, since the other reviewers either specifically requested this change or have not requested further clarifications.

Finally, the authors improved the introduction, but one sentence is still unclear (page 2, lines 30 to 35). I would write: "because fertility declines faster with age AND REQUIRES A LARGER PHYSIOLOGICAL COST FOR women than men, men are hypothesized to show stronger preferences for PHYSICAL cues of reproductive capacity (e.g., youth, health, and good nutritional status) in women than women do when assessing the attractiveness of potential mates".

We have made this change on page two: "First, because fertility declines faster with age and requires a larger physiological cost for women than men, men are hypothesized to show stronger preferences for physical cues of reproductive capacity (e.g., youth, health, and good nutritional status) in women than women do when assessing the attractiveness of potential mates [1]."

Reviewer 4's comments

The authors have been responsive to suggestions, and the current research proposal is acceptable.

Thank you very much.

Appendix D

Dear Chris,

Apologies that we missed this critical requirement for Stage Two submissions. This is our lab's first stage two submission, so we are still finding our feet, I'm afraid. We have added the following statement to the start of our Methods (page five): "The date of principle acceptance for this work was 16th October 2018. The accepted protocol is archived at <https://psyarxiv.com/sybp4/> (version one). Data and analysis code are archived at <https://osf.io/rkstx/>."

Best wishes, Ben Jones

Appendix E

Reviewer 2

The authors adhered to the registered experimental procedures, except for the number of participants recruited (higher than planned). The authors are honest and open about this mistake, and I do not think this is a major issue for the paper. However, I think it would be a nice addition for the paper if the authors could show that the results are the same when using the pre-registered number of participants (by randomly removing some participants in each group to match the registered experimental procedure).

We discussed this among the authors extensively prior to our submission of the initial version and decided against it. The data and analysis code are open so other researchers can carry out such analyses if they wish to.

One missing element is any attempt to show that the two groups being compared (UK and Chinese) are actually comparable. I know that the authors did not collect any demographic information apart from age (which is an issue when comparing cross-cultural groups), but the authors could at least make sure that there is no significant age difference between the groups being compared (as age can change mating preferences).

We include age in the open dataset so that other researchers can test predictions about age that were not included in our confirmatory analyses, where age was not included as a covariate. We also note that the decision to include a covariate such as age in an analysis should not be contingent on whether there is a significant difference between groups. Modeling shows that such a strategy decreases power (Mutz and Pemantle, 2013; <http://janhove.github.io/reporting/2014/09/26/balance-tests>).

I have one concern about the statistical analyses of the explanatory analysis. The authors are not controlling for multi-testing as they should, and some of their results would not be significant after a simple Bonferroni correction. For example (but not limited to): effect of participant sex in the Chinese sample for creativity and the interaction sex/relationship context for kindness in the UK sample.

We now discuss this point about our exploratory analysis. (page 16: “The results of these exploratory analyses should be treated cautiously, however, since many of the effects would not survive correction for multiple comparisons and may then be false positives.”)

There is one result that is in contradiction with the previous literature (and with the theory), that the authors are not discussing at all: it seems that male participants prefer an older partner than themselves for a long-term relationship (unless I misunderstood the methods). Given that this result is surprising, I think that one sentence explaining or at least acknowledging it would be useful.

We agree this is a surprising result (we have triple checked it and it is not a coding error) and note this in our Discussion. (pages 16 and 17: Our

exploratory analyses of age preferences showed that women had stronger preferences for older mates than did men. This replicates a well-established pattern of results in the mate preferences literature [3,5]. However, the men in our study did (on average) express a preference for mates older than themselves, particularly for long-term relationships. This is a surprising result, since men typically prefer mates younger than themselves [3,5]. Whether or not this is a pattern that replicates in similar samples (e.g., university students) is a question for future research.”)

The authors could add the number of participants (after exclusion) in each cell of the 6 tables presenting the results to increase clarity.

This information is given in the text, so we think would be redundant if added to the tables.

I really like the plots showing the distribution of scores. I was wondering if it would be possible to find a way to add the different means on the plots without diminishing their readability. I would also like to see the same graphs for the other variables (creativity, kindness, liveliness and maybe age) in the supplementary.

We have added the additional plots to the supplemental materials.

Reviewer 3

I got a bit confused by some of the used sub-headlines. More precisely, the authors use “Testing Prediction 1” on page 8 and page 11. The same is true for testing the other predictions. The authors might want to change one of the sub-headlines.

We have clarified that these refer to the analysis plan and results, respectively.

The exclusion rate seems rather high (especially for testing prediction 1). Can the authors give more information on how many participants got excluded for which reasons (e.g. how many failed the manipulation check, how many were outliers etc.)?

The high exclusion rate is a consequence of translation checks. We have added the requested information to our supplemental materials.

Regarding the exclusion rate again, the authors could think about computing robustness checks including all participants to see if there are any differences in results.

We disagree that this would be appropriate, since our exclusion criteria were specified in the analysis plan accepted at Stage One.

Please report effect sizes and/or confidence intervals for all effects.

We have added effect sizes to the full results in our supplemental materials.

Regarding the exploratory results section, can you add a sentence (to all results reported here) about the direction of the effect, as you did in the main analyses?

We have done so.

Do I understand it correctly, that men preferred slightly older women for long-term relationships (Table 6)? If yes, this is surprising. The authors may want to mention/ discuss this finding.

We agree this is a surprising result (we have triple checked it and it is not a coding error) and note this in our Discussion. (pages 16 and 17: Our exploratory analyses of age preferences showed that women had stronger preferences for older mates than did men. This replicates a well-established pattern of results in the mate preferences literature [3,5]. However, the men in our study did (on average) express a preference for mates older than themselves, particularly for long-term relationships. This is a surprising result, since men typically prefer mates younger than themselves [3,5]. Whether or not this is a pattern that replicates in similar samples (e.g., university students) is a question for future research.”)